# Targeting *KRAS4A* splicing through the RBM39/ DCAF15 pathway inhibits cancer stem cells

Wei-Ching Chen[1,6], Minh D. To [1,6], Peter M. K. Westcott[1,2,6], Reyno Delrosario[1], Il-Jin Kim[3], Mark Philips [4], Quan Tran[1], Saumya R. Bollam [1], Hani Goodarzi [5], Nora Bayani[1], Olga Mirzoeva [1] & Allan Balmain [1,5✉]

The commonly mutated human *KRAS* oncogene encodes two distinct KRAS4A and KRAS4B proteins generated by differential splicing. We demonstrate here that coordinated regulation of both isoforms through control of splicing is essential for development of *Kras* mutant tumors. The minor *KRAS4A* isoform is enriched in cancer stem-like cells, where it responds to hypoxia, while the major *KRAS4B* is induced by ER stress. *KRAS4A* splicing is controlled by the *DCAF15/RBM39* pathway, and deletion of *KRAS4A* or pharmacological inhibition of *RBM39* using Indisulam leads to inhibition of cancer stem cells. Our data identify existing clinical drugs that target *KRAS4A* splicing, and suggest that levels of the minor *KRAS4A* isoform in human tumors can be a biomarker of sensitivity to some existing cancer therapeutics.

[1] UCSF Helen Diller Family Comprehensive Cancer Center, San Francisco, CA, USA. [2] MIT Koch Institute for Integrative Cancer Research, Cambridge, MA, USA. [3] Guardant Health, Redwood City, California, USA. [4] NYU Cancer Institute, NYU School of Medicine, New York, NY, USA. [5] Department of Biochemistry and Biophysics, University of California San Francisco, San Francisco, CA, USA. [6] These authors contributed equally: Wei-Ching Chen, Minh D. To, Peter M. K. Westcott. ✉email: allan.balmain@ucsf.edu

The *KRAS* gene is the most frequently mutated oncogene in human cancers, particularly in tumors of the pancreas, colon and lung, and has consequently been a major focus of cancer drug discovery for decades. However, in spite of these huge efforts, tumors carrying *KRAS* mutations remain among the most difficult to treat, largely because of development of drug resistance due to tumor cell plasticity and/or acquisition of new mutations. A major advance in targeting a specific mutant KRAS G12C protein found in a subset of lung adenocarcinomas was recently described[1], but methods for direct targeting of *KRAS* that would apply to most or all *KRAS* mutant cancers are presently lacking. In order to develop a more integrative view of the biology of *KRAS* signaling and identify fresh approaches to inhibiting the *KRAS* pathway in cancers, we used a combination of mouse and human genetic analyses to investigate the functions and cell type-specific expression of the two known proteins produced by the *KRAS* locus, in normal tissues and during tumor development.

*KRAS* undergoes alternative splicing of the last exon to generate two isoforms, KRAS4A and KRAS4B, which are identical except for the 22/23 amino acids at the carboxyl terminus required for post-translational modifications and intracellular trafficking[2]. Germline deletion of the mouse *Kras* gene results in embryonic lethality[3], but specific deletion of only *Kras4A*, achieved by deletion of exon 4A[4] or knock-in of *Kras4B* cDNA (*Kras4B*[KI] mice, henceforth *Kras4A*[−/−])[5], had no effect on viability, suggesting that the main developmental functions of *Kras* are mediated through the *Kras4B* isoform. It has previously been demonstrated that mice lacking only the *Kras4A* isoform are resistant to chemically induced lung cancer[6,7], in spite of the fact that *Kras4A* is expressed only in a subpopulation of normal and tumor cells. These data led to the proposal that *Kras4A* plays an essential role in tumor development, possibly through effects on a minor stem cell population[6]. Subsequent analysis of human cancer data from TCGA also suggested a more important role for *KRAS4A* in human cancer than had previously been appreciated, as some tumors expressed unexpectedly high levels of this "minor" isoform[8].

Here, we design a genetic approach to investigate the distinct functions of *Kras4A* and *Kras4B* in mouse models and human cancers. This analysis shows that each isoform is individually dispensable for normal mouse development, but that expression of both splice isoforms from the same *Kras* allele is required for initiation and growth of *Kras* mutant cancers. Disruption of splicing through the *DCAF15/RBM39* pathway represents a novel route to target the *KRAS* pathway. Since *KRAS4A* is enriched particularly in a sub-population of cells with stem cell properties, our data suggest a model in which coordination of the stem-progenitor cell transition in cancers may be achieved through regulation of *KRAS* mRNA splicing, identifying a potential vulnerability in *KRAS* mutant tumors.

## Results

### Coordinated expression of *Kras4A* and *Kras4B* is essential for mouse lung tumorigenesis

In order to compare the roles of *Kras4A* and *Kras4B* in lung tumorigenesis, we used the previously published strategy[5] to insert a *Kras4A* cDNA into the *Kras* locus, thus generating an allele that completely lacks expression of *Kras4B* (*Kras4B*[−/−] mice)(Supplementary Fig. 1A–C). *Kras4B*[−/−] homozygous mice were born at normal Mendelian ratios, but were infertile. Expression of Kras4A and Kras4B isoforms in normal tissues from these mice showed the expected patterns (Supplementary Fig. 1), and mouse embryonic fibroblasts (MEFs) isolated from E13.5 embryos showed robust EGF-induced activation of MapK and Akt, indicating that Kras4A and Kras4B are both activated in response to EGF in cells that lack the alternative isoform

(Supplementary Fig. 2). Both knockout strains were crossed over 6 generations onto the *FVB/N* background, and injected with urethane. No tumors were found in treated *Kras4A*[−/−] animals, in agreement with previously published results[6,7]. However, this experiment showed that *Kras4B*[−/−] homozygous mice were also highly resistant to lung tumor development (Fig. 1A–D). Of the 18 *Kras4B*[−/−] homozygous animals treated with urethane, 8 developed one or two extremely small tumors each (Fig. 1A, B), and sequencing of *Kras* codons 12, 13, and 61 revealed no mutations. Thus, the effect of eliminating *Kras4B*, similar to loss of *Kras4A*, is to confer resistance to chemically-induced lung tumours with *Kras* mutations. Resistance is not due to changes in gene structure or absence of introns in the introduced cDNA construct, as mice carrying a similar cDNA encoding *Hras* rather than *Kras* isoforms developed lung cancers in which the cDNA construct carried the appropriate carcinogen-specific mutation[6].

Any tumors that arose in either *Kras4A*[+/−] or *Kras4B*[+/−] heterozygous mice carried mutations in the fully functional wild type allele (Supplementary Table 1), suggesting that the coordinated expression of both isoforms from the same *Kras* allele may be essential for tumorigenesis. To test this possibility, we carried out carcinogenesis studies in the lung using double heterozygous *Kras4A/4B* mice expressing both isoforms, but in which splice regulation is disrupted. Specifically, 1 double heterozygote was treated with 5 doses of urethane, 2 with 3 doses of urethane, and 3 with 3 doses of the carcinogen N-methyl-N-nitrosourea (MNU) in an attempt to increase the mutation burden and likelihood of generating mutant forms of one or both isoforms. No tumors were detected in the lungs of any of these double heterozygous animals 30 weeks after treatment (Fig. 1E). The resistance of mice expressing both isoforms, but on different alleles, suggests that the initiating *Kras* mutation has to be in an endogenous *Kras* gene capable of generating mutant versions of both splice variants.

### Both *KRAS4A* and *KRAS4B* contribute to the tumorigenicity of *KRAS* mutant human cancers

To assess the functional relevance of the oncogenic *KRAS* isoforms in human cancer cell lines in vitro, we carried out shRNA-mediated knockdown of each isoform in *KRAS* mutant cancer cell lines A549, SUIT2, YAPC, and H358. We screened several hybridoma cell lines to identify an antibody (10C11E4) capable of detecting endogenous KRAS4A (see Methods) which is normally expressed at low levels in cultured cells. Specific knock-down by isoform-specific shRNA was confirmed by Westerns with the KRAS4A-specific antibody 10C11E4, and the KRAS4B-specific antibody (Santa Cruz Kras2b antibody) (Supplementary Fig. 3A). Knockdown of either *KRAS4A* or *KRAS4B* in SUIT2 cells significantly compromised their capability to form colonies in soft-agar assays (Supplementary Fig. 3B). Reduction of *KRAS4A* or *KRAS4B* in A549 and YAPC cells has minor effects on growth in vitro, but significantly inhibited growth in vivo in immunocompromised mice (Supplementary Fig. 3B, C), suggesting, in agreement with the mouse studies, that both *KRAS4A* and *KRAS4B* are required for growth.

We next used CRISPR/Cas9 technology to generate complete knockouts of *KRAS4A* or *KRAS4B* in human *KRAS* mutant cells. Single guide RNAs (sgRNAs) were designed to target the Cas9 nuclease to *KRAS* exons 4A or 4B in the human lung and pancreatic cancer cell lines A549 (G12S mutation) and SUIT2 (G12D mutation). Frameshift insertions and deletions in *KRAS* exon 4A or 4B were determined by allele-specific cloning and sequencing of selected clones (Supplementary Fig. 4). We were unable to isolate any homozygous *KRAS4B* frameshift mutant clones, and noted a greatly reduced number of colonies proliferating beyond the first few cell divisions in this arm of the experiment.

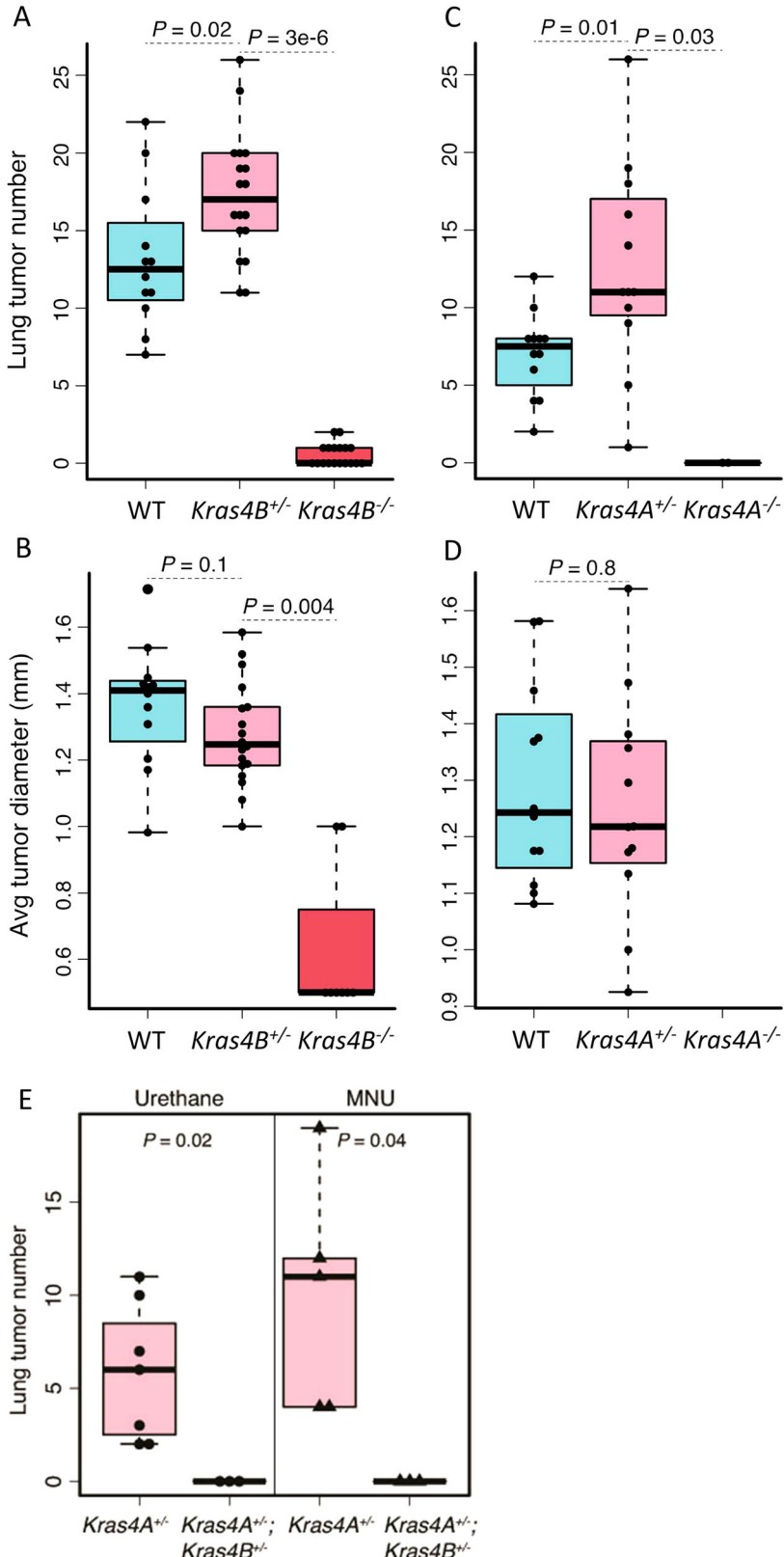

Complete loss of KRAS4A and reduced levels of KRAS4B were shown using isoform-specific antibodies (Fig. 2A). To determine the effects of *KRAS* isoform-specific knockouts on cell growth, cell proliferation on plastic and in soft agar were assessed. Homozygous knockout of *KRAS4A* and heterozygous knockout of *KRAS4B* significantly reduced growth in both assays (Fig. 2B–D).

Subcutaneously injected knockout cells in nude mice also had impaired tumor forming capacity compared to the parental SUIT2 or A549 cells (Fig. 2E). Taken together, these data indicate that *KRAS4A* and *KRAS4B* are both required for optimal growth of *KRAS* mutant lung and pancreatic tumor cells in vitro and in immunodeficient mice, but the effects of *Kras4a* inhibition on

**Fig. 1 *Kras4A* and *Kras4B* are essential for development of carcinogen-induced lung tumours.** Urethane-induced lung tumor number (**A**) and average diameter (**B**) in *WT*, *Kras4B*⁻ heterozygous and homozygous mice. Homozygous mice ($n = 18$) showed a highly significant reduction in tumor number and size, while heterozygous mice ($n = 18$) developed significantly more tumors than *WT* mice ($n = 12$). (**C**, **D**) Urethane-induced lung tumor number (**C**) and average diameter (**D**) in *WT* ($n = 12$), *Kras4A*⁻ heterozygous ($n = 12$) and homozygous mice ($n = 3$) showed very similar patterns as seen in the *Kras4B*⁻ cross. **E** Urethane-induced (left) and MMU-induced (right panel) lung tumor number in *Kras4A*$^{+/-}$ heterozygous ($n = 7$ in left panel and $n = 5$ in right panel) and double heterozygous *Kras4A/4B* mice ($n = 3$). No tumors were found in double heterozygous *Kras4A/4B* mice. All lung tumorigenesis experiments were performed on a *FVB/N* background. n is the individual mouse number. The center line is the median, the bottom of the box is the 25th percentile boundary, the top of the box the 75th, and the whiskers define the bounds of the data that are not considered outliers, with outliers defined as greater/lesser than ± 1.5 x IQR, where IQR = inter quartile range.

tumor growth in the context of an intact immune system remain to be established.

**Splicing of *KRAS4A* is regulated by the RBM39 RNA-binding protein.** Since our mouse genetic data suggested that controlled expression of both *Kras4A* and *Kras4B* isoforms from the same allele is necessary for tumor formation, we explored the possibility that small molecule inhibitors of different components of the splice site machinery may impact growth of *KRAS* mutant tumour cells[9,10]. We tested four inhibitors for effects on *KRAS4A/B* splicing: Pladienolide, which targets the splicing factor SF3B, disrupting the early stage of splice complex assembly[11]; Isoginkgetin, a bioflavonoid and general splicing inhibitor[12]; and two related sulfonamides Indisulam and Tasisulam, recently shown to cause proteasomal degradation of the RBM39 RNA binding protein and inhibition of *RBM39*-mediated splicing[13,14]. Isoginkgetin had no obvious effect on levels of *KRAS4A* or *KRAS4B* mRNA, while Pladienolide downregulated both *KRAS4A* and *KRAS4B* splice variants in SUIT2 cells (Fig. 3A, B) in agreement with its general role in disruption of the splicing machinery. Treatment with Indisulam or Tasisulam, specifically downregulated *KRAS4A*, but had no obvious effect on *KRAS4B* (Fig. 3A, B). Similar results were obtained using two additional cell lines A549 (Fig. 3C, D), and AsPC1 (Supplementary Fig. 5A, B), suggesting that *RBM39* controls *KRAS4A* levels. The downregulation of KRAS4A, but not KRAS4B, was also confirmed at the protein level in SUIT2 and A549 cells (Fig. 3E, F and Supplementary Fig. 5C), and in AsPC1 cells (Supplementary Fig. 5D).

To test the involvement of RBM39 in control of *KRAS* splicing, we first verified that Indisulam treatment led to downregulation of RBM39 in SUIT2 and A549 cells (Fig. 3E, F). In all cases where KRAS4A expression was reduced at RNA and protein levels, the RNA binding protein RBM39 was also reduced (Fig. 3E, F and Supplementary Fig. 5D). This effect was seen in all three cell lines for both Indisulam and Tasisulam, which target RBM39 through interaction with DCAF15[13]. The general splice inhibitor Pladienolide again caused inhibition of both 4A and 4B isoforms (Fig. 3E, F).

We then generated *RBM39* knockdown cells using CRISPRi/dCas9-KRAB[15] using specific guide RNAs targeting the *RBM39* gene in SUIT2 and A549 cells. Depletion of RBM39 caused a significant reduction in KRAS4A splice variant level but had no effect on KRAS4B (Fig. 3G), attesting to the specificity of *RBM39* in control of *KRAS* pre-mRNA splicing. Although the *RBM39* guide RNAs had a significant effect on RBM39 protein in A549 cells (Fig. 3G), the effect at the mRNA level was more modest (Supplementary Fig. 5E). It is possible that this reflects a feedback activation of *RBM39* transcription when the protein is depleted, as Indisulam treatment surprisingly caused a significant upregulation in *RBM39* mRNA while decreasing the protein levels (e.g. compare Fig. 3G and Supplementary Fig. 5E). Further studies of this apparent feedback control of *RBM39*, may shed light on the exact mechanisms involved.

Indisulam was originally proposed to act as an inhibitor of *CA9* (carbonic anhydrase 9) which is involved in *HIF1A* regulation

and hypoxia[16]. However CRISPRi/dCas9-KRAB downregulation of *CA9* mRNA (Supplementary Fig. 5F) had no effect on levels of KRAS4A, or on the impact of Indisulam on KRAS4A levels (Fig. 3G). While *RBM39* influences splicing of many pre-mRNAs genome-wide, these data support the proposal[13] that *RBM39* has a more limited set of physiological targets than would be expected for a general inhibitor of the splice machinery, and suggest that *KRAS4A* splicing may be controlled by *RBM39*. In support of this possibility, analysis of *RBM39* binding sites genome-wide identified several PAR-CLIP tags within the *KRAS* gene[17]. The most significant of these tags was located within *KRAS* close to exon 4A (Supplementary Fig. 6), and therefore represents a candidate target site for *RBM39* that may control *KRAS4A* splicing. However further detailed analysis by site-specific mutagenesis would be required to verify that *RBM39* influences *KRAS4A* splicing directly rather than through an intermediate protein complex.

In order to determine whether Indisulam can mimic the effect of *KRAS4A* deletion on human tumor growth in immunocompromised mice, we injected SUIT2 and A549 cells into NSG immunodeficient mice and treated the animals carrying established tumors with Indisulam by retro-orbital injection (Fig. 3H). Both cell lines showed significant tumor growth inhibition by Indisulam in vivo, although the effect was more pronounced for SUIT2 cells (Fig. 3H). Others have also demonstrated a significant effect of Indisulam on growth of the *KRAS* mutant HCT116 human colon cancer cell line in vivo[13].

Our data therefore identify an existing druggable pathway involving inhibition of *RBM39* that can target expression of the minor *KRAS4A* isoform. Interestingly, inspection of the DEP-MAP portal shows that Indisulam activity in human tumor cell lines is highest in a subset of tumors derived from tissues such as the bone, haematopoietic system, soft tissues, and central nervous system. Ranking of these tumor cell lines according to the ratio of *KRAS4A* and *KRAS4B* levels in fact shows that these tumor types have the lowest levels of this ratio (Fig. 3I, J and Supplementary Fig. 7A). To establish whether there is a significant effect of *KRAS4A* on Indisulam sensitivity, we obtained several of the cell lines characterized in DEPMAP and repeated cell growth assays in the presence of Indisulam, and also directly measured the levels of expression of the *KRAS* isoforms. We chose 2 blood cancer cell lines (Jurkat and HL60, wild-type *KRAS*), 2 lung cancer cell lines (NCI-H661 and NCI-H1703, wild-type *KRAS*), and 2 pancreatic cancer cell lines (YAPC and SW1990, mutant *KRAS*) for these studies. Supplementary Fig. 7B shows that blood cancer cells are more sensitive to Indisulam than lung NCI-H661 and pancreatic cancer cell lines, in agreement with the results from DEPMAP analysis. YAPC showed the highest KRAS4A level and HL60 cells have a lower KRAS4A/KRAS4B ratio than lung and pancreatic cancer cells (Supplementary Fig. 7C), in agreement with the results from the Cancer Cell Line Encyclopedia (CCLE). Treatment with Indisulam downregulated RBM39 and KRAS4A, but not KRAS4B, in Jurkat and HL60 cells (Supplementary Fig. 7D). Compared to the effect of Indisulam in SUIT2,

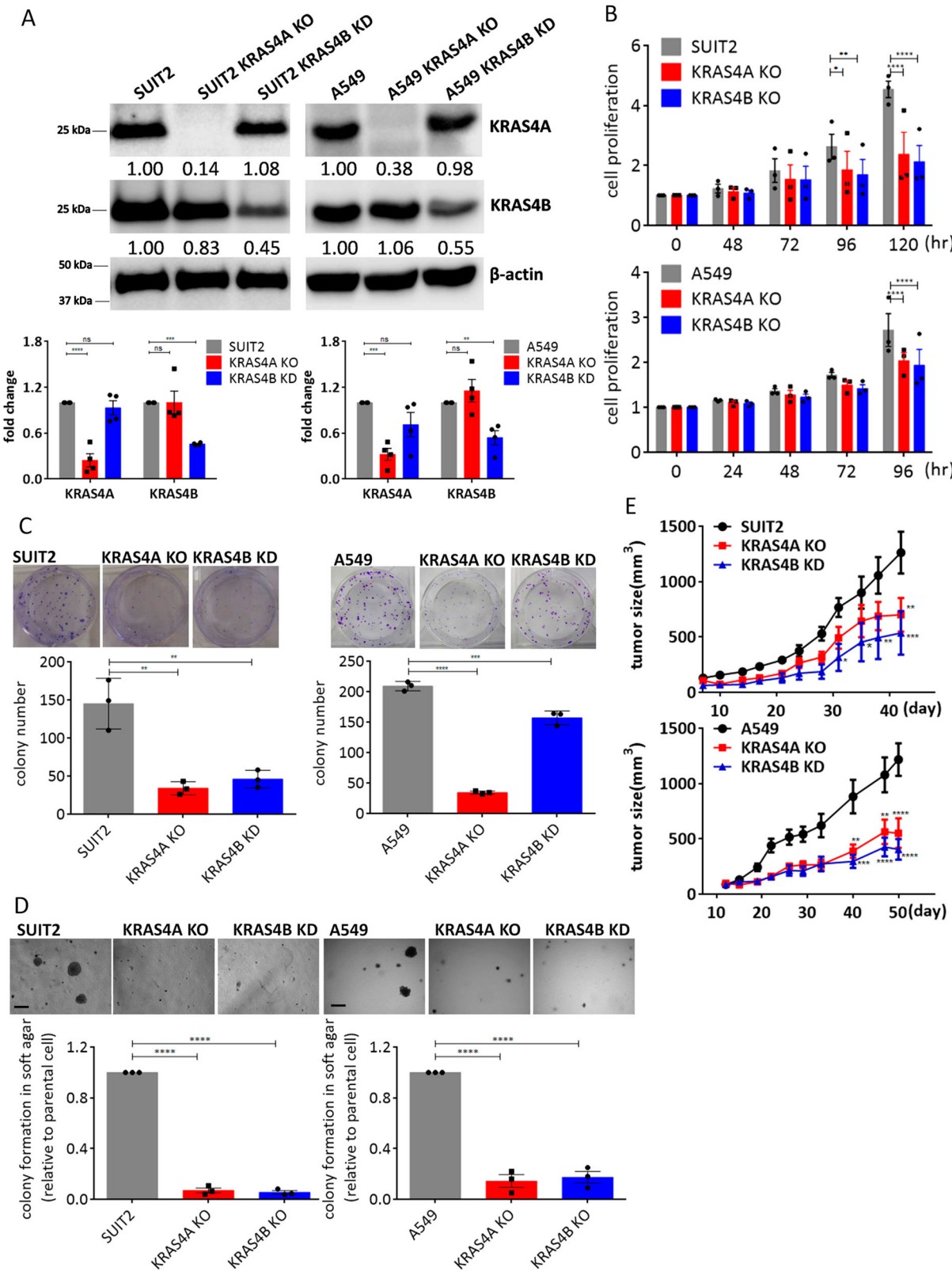

downregulation of KRAS4A by Indisulam was seen at lower concentration in HL60 (Supplementary Fig. 7E). We also looked at the sensitivity of tumor cells with WT *KRAS* to treatment with Indisulam, and demonstrated that KRAS4A was downregulated in response to both Indisulam and knockdown of RBM39 in H1650 cells (Supplementary Fig. 7F). We therefore propose that the absolute level of *KRAS4A*, or the ratio of *KRAS4A* to *KRAS4B*,

may be a biomarker of sensitivity to Indisulam or other drugs that interfere with *KRAS4A* splicing, but additional clinical studies will be necessary to confirm this possibility.

**KRAS4A is enriched in cells with cancer stem cell properties.** The above results showing that inhibition of *KRAS4A* can impact

**Fig. 2 Effects of loss or reduced expression of KRAS4A or 4B on growth of human cancer cells in vitro and in vivo. A** Western blotting using isoform-specific antibodies showed complete loss of KRAS4A (*KRAS4A* KO) in A549 and SUIT2 cells. KRAS4B is reduced but not eliminated in cells heterozygous for CRISPR/Cas9-mediated heterozygous loss of *KRAS4B* (*KRAS4B* KD). Lower panel: Quantification of KRAS4A and KRAS4B protein levels using imageJ from $n = 4$ independent experiments confirmed the specificity of the sgRNAs used for the CRISPR knockouts. Data are presented as mean ± s.e.m. **$P <$ 0.01; ***$P < 0.001$; ****$P < 0.0001$ by two-way ANOVA with Sidak's multiple comparisons test. CRISPR-Cas9 induced knockouts of *KRAS4A* and *KRAS4B* in SUIT2 and A549 cells showed reduced growth in vitro on plastic (**B**), colony formation (**C**), and soft agar (**D**). Data are presented as mean ± s.e.m from $n = 3$ indeoendent replicates. *$P < 0.05$; **$P < 0.01$; ***$P < 0.001$; ****$P < 0.0001$ by two-way ANOVA with Bonferroni's multiple comparison for (**B**) and one-way ANOVA with Tukey's multiple comparison for (**C**) and (**D**). The scale bar is presented as 500 µm. **E** Growth of homozygous *KRAS4A* KO and heterozygous *KRAS4B* KD SUIT2 and A549 cells after subcutaneous injection into nude mice. Reduction in expression of both isoforms caused a significant decrease in tumor forming capacity in vivo. Data are presented as mean ± s.e.m. $n = 10$ mice for parental cells and $n = 5$ mice for *KRAS4A* KO or *KRAS4B* KD cells. *$P < 0.05$; **$P < 0.01$; ***$P < 0.001$;****$P < 0.0001$ by two-way ANOVA.

tumor growth in vivo was surprising, in view of the fact that *KRAS4A* levels are generally much lower than those of the major isoform *KRAS4B*, and *Kras4A* is only expressed in a subpopulation of cells in normal tissues or tumors[6,18]. We tested the hypothesis that *KRAS4A* may be preferentially expressed in a subpopulation of cancer stem cells[19,20] using a well characterized assay for "side population" cells[21] that have been shown to be enriched in stem cell properties. We isolated side population cells and analyzed sphere formation efficiency of side population and non-side population cells from 3 different cell lines (Supplementary Fig. 8A, B). Expression of a marker of side population cells, (*ABCG2*)[22] was significantly elevated in side population cells from both A549 and SUIT2 cells, as well as from an additional pancreatic *KRAS* mutant cell line AsPC1 (Fig. 4A). Importantly, *KRAS4A* expression was also enriched in the same stem cell populations, as shown by specific TaqMan analysis of both *KRAS* splice isoforms (Fig. 4B). Homozygous knockout cells that no longer expressed *KRAS4A* had a reduced proportion of side population cells, which could be rescued by re-introduction of a functional *KRAS4A* expression construct (Fig. 4C). The activity of the side population stem cell marker aldehyde dehydrogenase (ALDH)[23], was also reduced in the *KRAS4A* knockout cells, and restored by the re-introduced *KRAS4A* isoform (Fig. 4D). In contrast to these observations regarding *KRAS4A*, there was no significant difference in *ALDH* expression in cells with reduced levels of *KRAS4B* in the heterozygous knockout lines (Fig. 4D) indicating that expression of *KRAS4A*, but not *KRAS4B* is linked to cells with stem cell properties. In an attempt to increase the specificity for isolation of side population stem cells, we used additional reported markers of stem cells (CD166 and CD133) in combination with ALDH for FACS analysis. CD133 provided some additional enrichment in SUIT2 cells but not in A549 cells, while CD166 did not improve discrimination between side population and non-side population cells in either cell line (Supplementary Fig. 8C, D). Finally, in accordance with the reduction in *KRAS4A* by guide RNAs targeting *RBM39* (Fig. 3), inhibition of *RBM39* also reduced the proportion of ALDH positive cells in both SUIT2 and A549 cells (Supplementary Fig. 5G).

**KRAS4A and KRAS4B are induced by different types of cellular stress.** The observation that both *KRAS* splice isoforms are required for tumor growth suggested that they may have different functions in cancer stem and progenitor cells. We therefore investigated the possibility that *KRAS4A* and *KRAS4B* may be responsive to different kinds of cellular stress. Treatment of parental A549, SUIT2, and AsPC1 cells with $CoCl_2$ induced an increase in expression of *HIF-1α*, a known marker of hypoxia (Fig. 5A). *KRAS4A*, but not *KRAS4B* (Fig. 5B) showed a significant increase in expression as a consequence of this treatment, and the increase was predominantly in the side population cells rather than in the bulk cell population (Fig. 5C). Rapid tumor

growth can lead to a hypoxic state which has been linked to stem cell activation and metabolic reprogramming[24]. In contrast, Endoplasmic Reticulum (ER) stress, which has been identified as a therapeutic target in *KRAS* mutant cancers[25] caused a significant increase in the ER marker *HSPA5* (also known as *GRP78*) and in *KRAS4B*, but not *KRAS4A*, in both A549 and SUIT2 cancer cell lines (Fig. 5D, E). Although both $CoCl_2$ and tunicamycin may not be completely specific for hypoxia and ER stress respectively, these data emphasize the functional differences between the two isoforms in responding to different types of stress in the same human cancer cell lines.

**Low levels of KRAS4A are associated with increased expression of cell cycle/mitotic markers.** We next investigated the relative expression levels of the two *KRAS* splice isoforms in *KRAS* mutant human lung cancer data sets[26,27]. We assessed the isoform-specific expression of *KRAS* in a cohort of 86 human lung adenocarcinomas[26] using TaqMan probes designed to recognize *KRAS4A* or *KRAS4B* transcripts. Twenty three of these tumors harbor activating mutations in *KRAS* codons 12 or 13, while the remaining 63 are WT at *KRAS* codons 12, 13, and 61. TaqMan analysis revealed that both total *KRAS* and *KRAS4A* expression were significantly elevated in the *KRAS* mutant subset of tumors, while *KRAS4B* expression showed no significant difference (Fig. 6A–C). As gene expression microarray data were available for these samples, we separated them into two subgroups according to their *KRAS4A/KRAS4B* transcript ratios (high ($n = 8$) or low ($n = 7$)). Supplementary Table 2 shows the top 100 genes that are highly expressed in samples with a low *KRAS4A/KRAS4B* transcript ratio. Surprisingly, in view of the data in Fig. 2 indicating that deletion of *KRAS4A* in human carcinoma cells in vitro causes some growth inhibition, GO enrichment analysis of genes highly expressed in primary lung carcinomas with low *KRAS4A* levels identified pathways linked to the cell cycle, microtubule organization, mitosis, and DNA damage responses (Supplementary Table 3). Gene set enrichment analysis confirmed enrichment for cell cycle genes in the low *KRAS4A/KRAS4B* samples (Fig. 6D). The reasons underlying this difference are unclear and may be due to differences in activation of proliferation and DNA damage checkpoint genes in vitro and in vivo, or to fundamental differences in growth control including the role of an active immune system in human patient samples. These questions will be the subject of future investigations.

We also analyzed the isoform-specific expression levels of *KRAS* from TCGA lung adenocarcinoma (LUAD) samples. Although RNA was not available for isoform-specific TaqMan analysis, RNAseq reads spanning the alternatively spliced exon junctions of *KRAS* were used to estimate the percentage of total *KRAS* transcripts made up by *KRAS4A* (see Methods). Highly significant increases in both *KRAS4A* and *KRAS4B* transcripts were found in RNAseq data from *KRAS* mutant versus WT lung adenocarcinoma (LUAD) tumors, but *KRAS4A* showed a greater

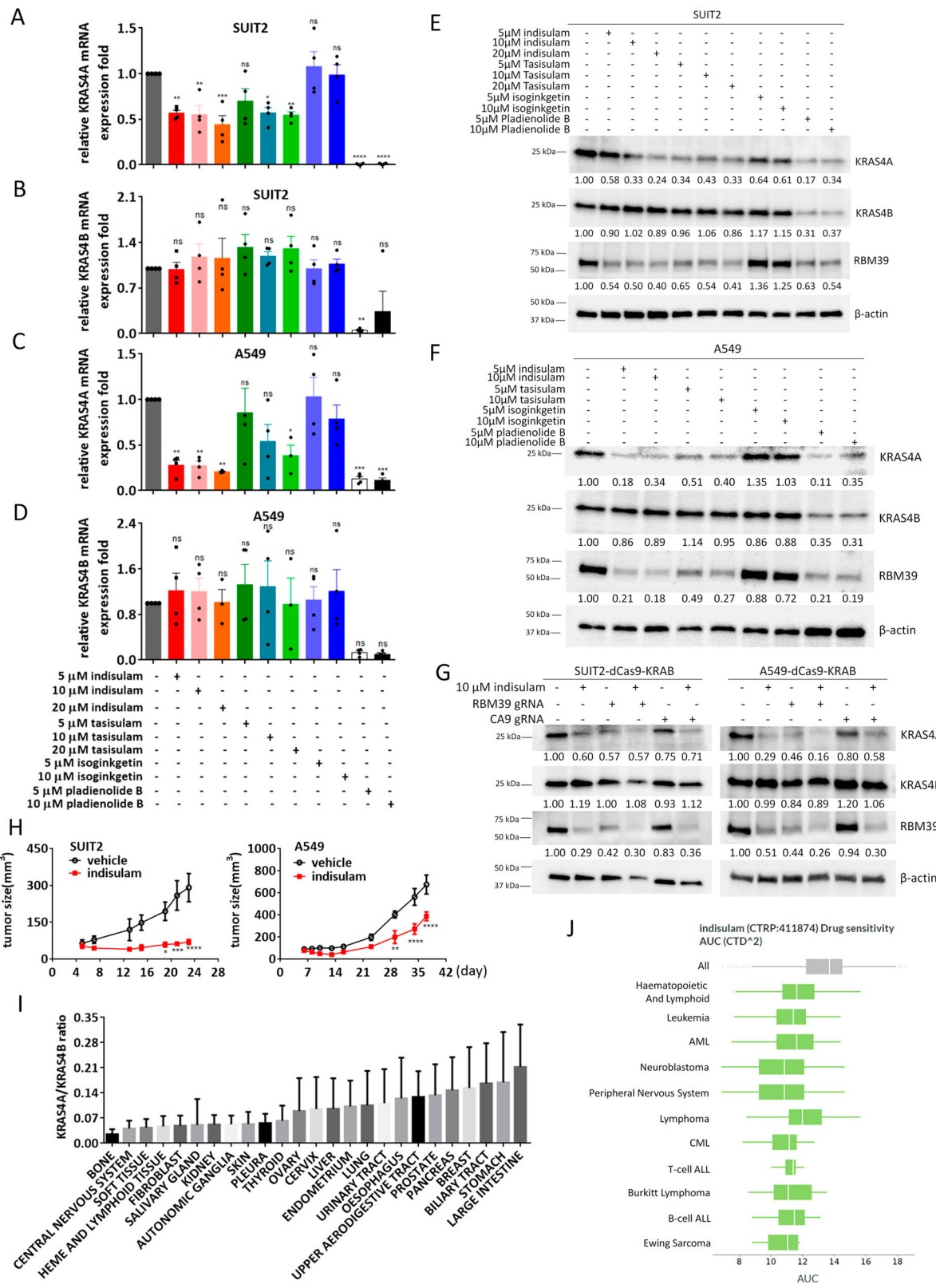

fold increase (1.7) than *KRAS4B* (1.2) (Supplementary Fig. 9A±C). This analysis confirmed that *KRAS* mutant lung tumors on average have a significantly higher ratio of *KRAS4A* to *KRAS4B* than *KRAS* WT tumors. Similar results were recently reported by Stephens et al[28]. To determine if splicing shifts towards higher *KRAS4A* expression in other cancer types, these analyses were extended to the TCGA pancreatic adenocarcinoma (PAAD) and

colorectal adenocarcinoma (COADREAD) RNAseq datasets[29]. Significant increases of *KRAS4A* and *KRAS4B* expression in *KRAS* mutant tumors were observed, with greater fold increase in *KRAS4A* (Supplementary Fig. 9D–I). The percentage of total *KRAS* transcripts made up by *KRAS4A* showed similar increases in *KRAS* mutant PAAD and COADREAD, although significance was not reached, possibly due to the limited number of samples in

**Fig. 3 The RBM39 RNA-binding protein mediates _KRAS4A_ splicing.** (A-D) The _KRAS4A_ and _KRAS4B_ mRNA levels were assessed by TaqMan analysis in SUIT2 (**A**, **B**) and A549 (**C**, **D**) cells after small molecule inhibitor treatment for 48 hr. The specific inhibitors used and their concentrations are shown below the plots. Data are presented as mean ± s.e.m from $n = 4$ independent experiment. *$P < 0.05$; **$P < 0.01$; ***$P < 0.001$; ****$P < 0.0001$ by one-way ANOVA with Tukey's multiple comparison. **E, F** The KRAS4A, KRAS4B and RBM39 protein levels were assessed in SUIT2 (**E**) and A549 (**F**) by Western blotting after small molecule inhibitor treatment for 48 h. Quantification of KRAS4A, KRAS4B and RBM39 levels was carried out using imageJ software. **G** Two sgRNAs targeting _RBM39_ or _CA9_ were transfected into BFP + SUIT2 or A549 cells stably expressing dCas9-KRAB. Cells were incubated with small molecule inhibitors for 48 h and then analyzed by Western blotting. **H** In vivo inhibition of tumor growth of SUIT2 and A549 cells in immunocompromised mice. NSG mice ($n = 5$) were injected subcutaneously with SUIT2 cells or A549 cells and treated with indisulam (40 mg/kg) for 8 days (SUIT2) or 5 days (A549) by retro-orbital injection. Data are presented as mean ± s.e.m. *$P < 0.05$; **$P < 0.01$; ***$P < 0.001$; ****$P < 0.0001$ by two-way ANOVA with Bonferroni's multiple comparisons test. **I** CCLE summary of the ratio of KRAS4A to KRAS4B in human tumor cell lines from different tissues ($n = 6260$). Data are presented as mean ± s.d. (**J**). DEPMAP analysis of sensitivity to Indisulam across a wide range of cell lines from different human tumor types ($n = 719$ cell lines). The center line is the median, the bottom of the box is the 25th percentile boundary, the top of the box the 75th, and the whiskers define the bounds of the data. The detailed sample sizes are shown in the source data.

these cohorts with known _KRAS_ mutational status. Altogether, these data argue that the increase of _KRAS_ expression observed in _KRAS_ mutant cancers is driven not only by an overall increase in _KRAS_ expression, but by altered splicing favoring an increase in the ratio of _KRAS4A_ to _KRAS4B_ transcripts. In order to verify the observation in Supplementary Table 2 and 3 showing enrichment of mitotic/cell cycle pathways in UCSF lung tumors with a low _KRAS4A/KRAS4B_ ratio, we separated the available TCGA lung carcinoma samples into _KRAS4A/KRAS4B_ high ($n = 7$) and low ($n = 7$) groups and pancreatic adenocarcinoma samples into _KRAS4A/KRAS4B_ high ($n = 9$) and low ($n = 9$) groups and examined the total RNASeq data for differences in these gene expression pathways. Figure 6E, F and Supplementary Table 4 show the GO enrichment for pathways over-represented in the low _KRAS4A/KRAS4B_ ratio subgroup, again demonstrating that cell cycle/mitotic gene transcripts are enriched in tumors with low KRAS4A mRNA levels.

We further verified the proposed relationship between side population cells, stemness, and _KRAS4A_ levels using microarray data derived from gene expression analysis of human malignant pleural mesothelioma (GSE33734 dataset). In agreement with the observation that side population stem cells are mostly quiescent[30], expression of a cell cycle marker (_MCM7_) was decreased, while genes associated with growth suppression in stem cells (_GFI1_ and _NDN_) were increased in side population cells (Supplementary Fig. 10A–C)[22,31–33]. Importantly, a probe located within KRAS exon4A (ILMN_1652104) was also significantly increased in side population cells compared to non-side population cells (Supplementary Fig. 10D). _ABCG2_ expression was also enriched in side population cells (Supplementary Fig. 10E), in agreement with our data in Fig. 4A, B.

These data based on human tumors from TCGA suggested that low _KRAS4A_ levels in vivo may be linked to increased sensitivity to some drugs that target the G2/M checkpoint during cell cycle progression. Since rigorous testing of this hypothesis would require new human clinical studies, we first examined the possible relationship between expression of _KRAS4A_ and altered sensitivity to some existing therapeutic drugs using in vitro cell culture conditions. Rigosertib[34] and KG-5[35] target microtubules and are active in the G2/M cell cycle checkpoint, but also have other activities[36,37]. No differences were found in the effects of these drugs on growth of parental and _KRAS4B_ knockdown cells, but _KRAS4A_ knockout SUIT2 cells showed increased sensitivity to both compounds (Fig. 6G, H). These data support the proposal that inhibition of _KRAS4A_ can increase sensitivity to a subset of known cancer therapeutics in vitro, but further in-depth screening using small molecule libraries or CRISPR/Cas9 approaches will be required to identify vulnerabilities that may be dependent on variations in _KRAS4A/KRAS4B_ levels.

## Discussion

**Essential functions for both KRAS4A and KRAS4B in cancer development**. Targeting the _KRAS_ pathway for cancer therapy has been particularly challenging, in spite of the vast amount of knowledge accumulated on the genetic alterations leading to pathway activation in tumors, and on the signaling networks that are driven by activated RAS oncoproteins. The lack of success in direct inhibition of mutant RAS proteins led to a focus on downstream RAS effectors, and to identification of many targeted drugs that are approved or presently in clinical trials[38]. A major problem encountered in all targeted approaches to cancer therapy including those targeting KRAS, is development of drug resistance, variously attributed to development of novel mutations that circumvent the effects of drug exposure, or to stem cell plasticity resulting in a new cell fate with loss of dependence on the original driver mutation[39]. The data presented here help us to address these major issues by taking an integrative approach to understanding the biological functions of both proteins produced by the _KRAS_ locus, rather than focusing only on the more abundantly expressed isoform. A combination of mouse and human genetic approaches allowed us to identify a novel route to targeting of _KRAS_ at the expression level by inhibition of _KRAS4A_ splicing, and by demonstrating that the two splice variant proteins produced by the _KRAS_ locus are primarily expressed in distinct stem and progenitor cells. This cell heterogeneity is a cardinal feature of _KRAS_ mutant tumors19, 20, 38 contributing to the cell plasticity that impacts therapeutic responses and development of drug resistance.

The existence of two distinct isoforms of _Kras_ has been known for many years, but almost all of the research on _KRAS_ has focused on the _KRAS4B_ isoform which is widely and abundantly expressed across a range of tissues. Both _Kras4A_ and _Kras4B_ isoforms are required for initiation and/or progression of carcinogenesis in the mouse lung, as tumors are only induced in animals that have at least one functional _Kras_ allele capable of expressing both proteins. Double heterozygous _Kras4A/4B_ mice were also extremely resistant, even under conditions where the carcinogen dose was significantly increased, suggesting that coordinated splicing to generate mutant versions of both isoforms from the same allele is required for tumorigenesis.

In the mouse, _Kras4A_ is expressed during differentiation of pluripotent embryonic stem cells and in a subset of cells in adult tissues[18], raising the possibility that _Kras4A_ has specific functions in a small population of cells with stem cell properties[6]. In agreement with this possibility, human _KRAS4A_, but not _KRAS_4B, is enriched in stem cell-like side population cells derived from human cancer cell lines. Loss of _KRAS4A_, but not _KRAS4B_, causes a decrease in the proportion of cells with side population characteristics, as well as decreased activity of ALDH,

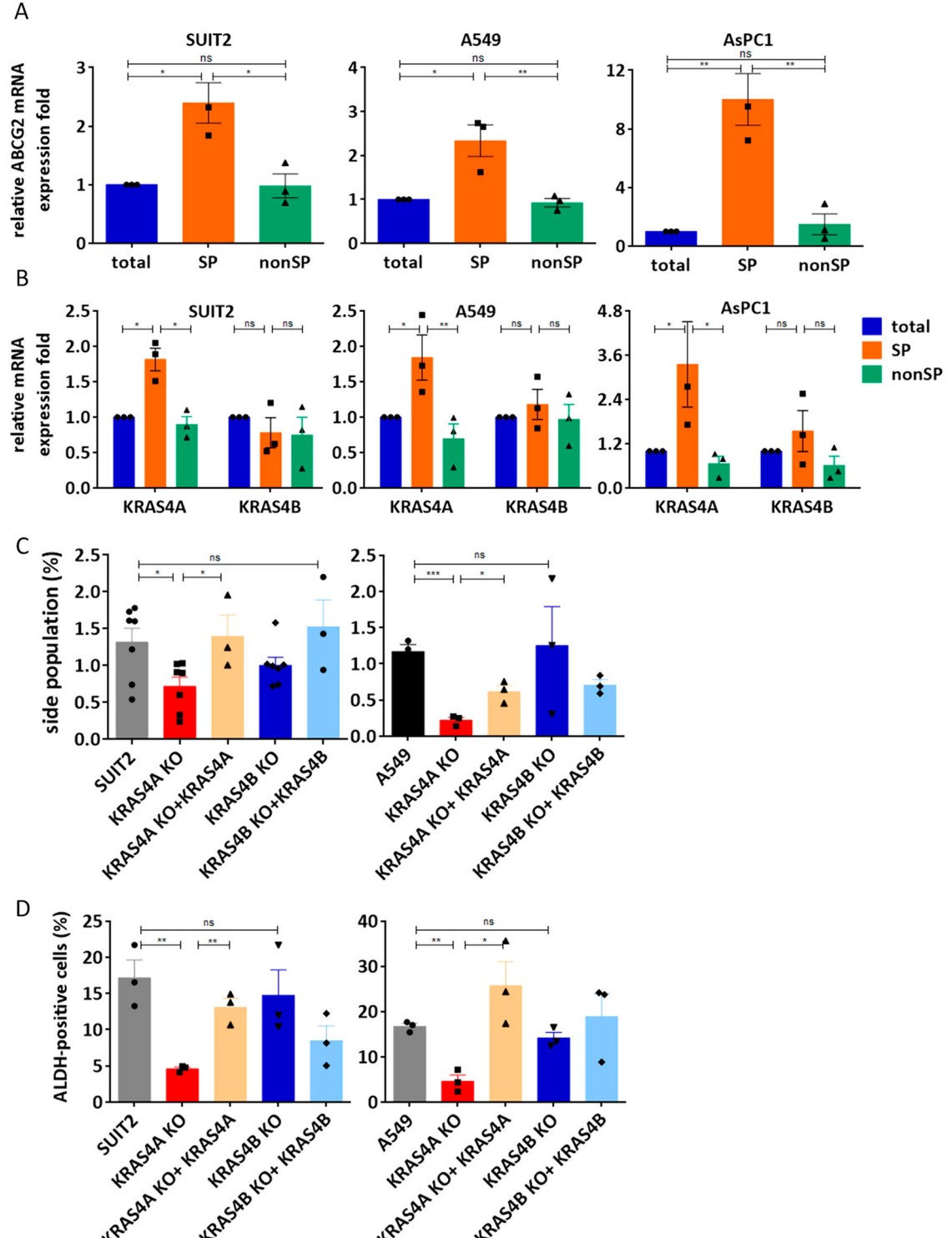

**Fig. 4 *KRAS4A* regulates stemness properties in human cancer cells.** Stem cell side populations (SP) were assessed in A549, SUIT2 and AsPC1 cells by Hoechst 33342 staining. **A** *ABCG2* levels were significantly increased in side population cells from all 3 cell lines. Data are presented as mean ± s.e.m from $n = 3$ independent experiments. *$P < 0.05$; **$P < 0.01$ by one-way ANOVA with Tukey's multiple comparison. **B** *KRAS4A*, but not *KRAS4B* was significantly increased in levels by TaqMan analysis of side population cells. Data are presented as mean ± s.e.m from $n = 3$ independent experiments. *$P < 0.05$; **$P < 0.01$ by one-way ANOVA with Tukey's multiple comparison. **C**, **D** Loss of *KRAS4A* reduces the proportion of side population cells (**C**) as well as the expression of the ALDH side population marker (**D**). Both the proportion of side population cells and ALDH expression levels were restored by transfection of a *KRAS4A* expression construct using fluorescence-activated cell sorting analysis. Data are presented as mean ± s.e.m from $n = 7$ independent experiments in left panel of (**C**) and $n = 3$ independent experiments in right panel of (**C**) and (**D**). *$P < 0.05$; **$P < 0.01$; ***$P < 0.001$ by Unpaired two-tailed *t*- test.

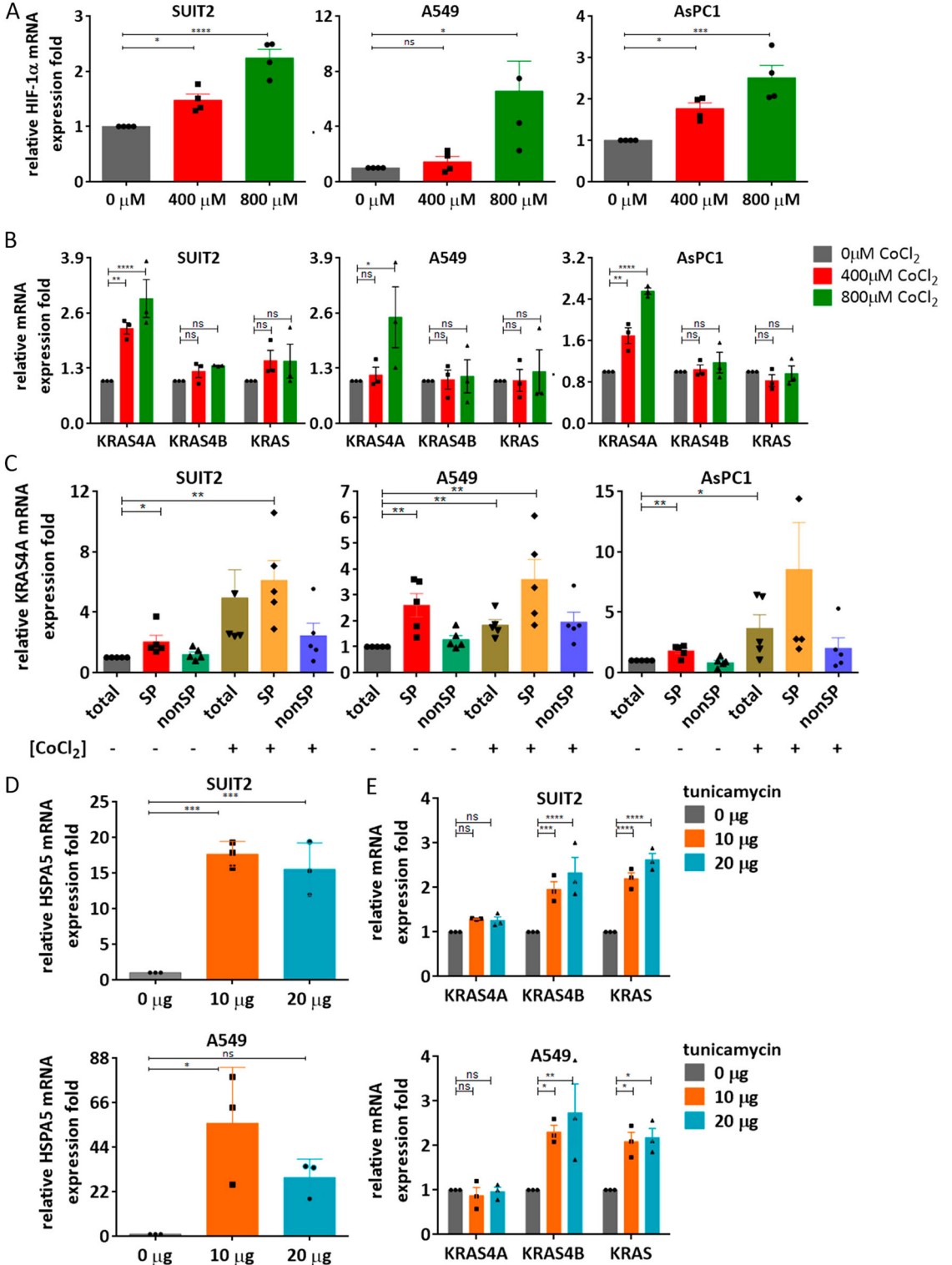

a known marker of these cells. In addition, hypoxic conditions which are known to lead to reactivation of stem cells, cause an upregulation of *KRAS4A*, but not *KRAS4B*. We propose that the splicing of *KRAS* to generate the 4A and 4B isoforms may be a critical event in controlling stress responses and the proliferative or metabolic requirements in stem and progenitor cells (Fig. 7). Indeed, cancer stem and progenitor cells have previously been shown to exhibit differences in their growth and metabolic propserties[20,40] and cancer stem cells have also been linked to development of drug resistance[41].

The sulfonamide drug Indisulam has been in various cancer clinical trials for several years but its specific targets were unknown until the recent demonstration that it can induce degradation of the RBM39 RNA-binding protein, by tethering to DCAF15[13,42]. The mechanisms and targets downstream from *RBM39* have been unclear, but we demonstrate here that an important consequence of *RBM39* loss is inhibition of the minor *KRAS4A* isoform. CRISPRi-mediated inhibition of *RBM39* reduces *KRAS4A* levels, as well as the proportion of side population stem cells in human cancer cell lines. While *RBM39*

**Fig. 5 Different types of cellular stress affect expression of _KRAS_ isoforms. A, B** The _HIF-1α_, _KRAS4A_, _KRAS4B_ and total _KRAS_ mRNA levels were assessed by TaqMan analysis in A549, SUIT2, and AsPC1 cells after CoCl$_2$ treatment for 48 hr. The hypoxia marker _HIF-1α_ was significantly increased by CoCl$_2$ in all 3 cell lines. Data are presented as mean ± s.e.m from $n = 4$ independent experiment. *$P < 0.05$; ***$P < 0.001$; ****$P < 0.0001$ by one-way ANOVA with Tukey's multiple comparison for (**A**). CoCl$_2$ increased _KRAS4A_, but not _KRAS4B_ levels in A549, SUIT2, and AsPC1 cells. Data are presented as mean ± s.e.m from $n = 3$ independent experiment. *$P < 0.05$; **$P < 0.01$; ****$P < 0.0001$ by two-way ANOVA with Bonferroni's multiple comparison for (**B**). **C** The side populations were isolated with or without CoCl$_2$ treatment and _KRAS4A_ expression level was determined in sorted side population and non-side population cells by TaqMan analysis. _KRAS4A_ was significantly increased by CoCl$_2$ in side population cells. Data are presented as mean ± s.e.m from $n = 5$ independent experiments. *$P < 0.05$; **$P < 0.01$ by unpaired two-tailed _t_-test. **D, E** _HSPA5_, _KRAS4A_, _KRAS4B_, and total _KRAS_ mRNA levels were assessed by TaqMan analysis in A549 and SUIT2 after tunicamycin treatment for 24 h. The ER stress marker _HSPA5_ was significantly increased by tunicamycin. Tunicamycin increased _KRAS4B_ and total _KRAS_, but not _KRAS4A_ levels. Data are presented as mean ± s.e.m from $n = 3$ independent experiments. *$P < 0.05$; **$P < 0.01$; ***$P < 0.001$; ****$P < 0.0001$ by one-way ANOVA with Tukey's multiple comparison for (**C**) and by two-way ANOVA with Bonferroni's multiple comparison for (**D**).

clearly has other splice targets in the transcriptome, these data link a druggable pathway with targeting of _KRAS_ expression.

These results have several possible clinical applications. The levels of _KRAS4A_, and/or the _KRAS4A/KRAS4B_ ratio, may be a biomarker for predicting responses to known cancer drugs. Evidence in favor of this possibility can be found in existing data sets, for example the DepMap and CCLE resources on cancer drug sensitivity across a large panel of human tumor cell lines. Levels of _KRAS4A_ and the _KRAS4A/KRAS4B_ ratio across this cell line panel are low in tumors derived from bone, neuronal, and haematopoietic/lymphoid tissues, but are significantly higher in epithelial tumors of the lung, pancreas, and intestine (Fig. 3). The haematopoietic, lymphoid, and neuronal tumor cell lines show the highest sensitivity to several cell cancer therapeutics including Rigosertib and Indisulam, although interpretation of these results is complicated by the off-target effects of these drugs, which have been reported to inhibit several targets including microtubules and cyclin-dependent kinases[43]. The present data, therefore, do not allow us to conclude that drug effects are due to low _KRAS4A_ levels, G2/M checkpoint inhibition, or a combination of both. The relationship between _KRAS4A_ levels, cell proliferation and DNA damage responses in vivo in the presence of an intact immune system also remains to be clarified.

Inhibition of _KRAS4A_ through interference with the RBM39 splice machinery[13], or directly by inhibition of splicing using oligonucleotide-based drugs[44], may reveal new vulnerabilities to existing drugs, as suggested by the genetic experiments in Fig. 6. Indisulam-mediated inhibition of _KRAS4A_ provides an important step in this direction, although this drug clearly has other targets and may have limited possibilities for combinatorial therapies. Further screening will be required to identify drug combinations that may optimally impact cancer stem-progenitor cell populations, leading to more effective treatments for _KRAS_ mutant tumors.

## Methods
**Mice**. The _Kras4B_ KO mice were generated as previously described[5]. Specifically, targeting strategy for the generation of a _Kras_ allele with _Kras4A_ cDNA knocked into the endogenous locus. _Kras4A_ cDNA spanning a portion of exon 2 through exon 4A and 3′ UTR was homologously inserted into the _Kras_ locus between exons 2 and 3. Chemical carcinogenesis of lung was performed by intraperitoneal injection of urethane or MNU as previously described[6]. Chemical carcinogenesis of the lungs by urethane was done in 6 week old male ($n = 33$) and female ($n = 40$) mice, and the mice were sacrificed 20 weeks later for tumor number analysis. Chemical carcinogenesis of the lungs in double heterozygotes mice was done in 6 week old male ($n = 11$) and female ($n = 7$) mice by urethane or MNU as the indicated dose in the Source data, and the mice were sacrificed 30 weeks later for tumor number analysis. The tumor size and number were assessed under a dissecting microscope with the aid of a ruler and reference images of a range of circles with different diameters. All tumor scoring was performed blind to mouse genotype.

All animal experiments were approved by the UCSF Institutional Animal Care and Use Committee IUCAC (AN102384-03D).

**Cell culture**. SUIT2, A549, and mouse embryonic fibroblasts (MEFs) isolated from E13.5 embryos were cultured in high glucose DME media (Gibco) containing 10% fetal bovine serum (Gibco) and 1% penicilin-streptamicine (Gibco). Human cell line SUIT2 was from AcceGen. A549, H358, and AsPC1 were from ATCC. All the cells were authenticated by STR profiling and were tested negative for mycoplasma contamination.

**Generation of _KRAS4A_, _KRAS4B_, and _RBM39_ knockout or knockdown cells**. Human _KRAS4A_ knockout and _KRAS4B_ knockdown cells were generated by CRISPR-Cas9-mediated genome engineering, as previously described[45]. sgRNA targets were GGAGGATGCTTTTTATACAT for _KRAS4A_ and TTCTCGAAC-TAATGTATAGA for _KRAS4B_. To confirm the insertions and deletions of each allele, the PCR products around the CRISPR-cas9 targeted sites were amplified from gDNA of established stable clones and cloned into pMiniT 2.0 vector (NEB), followed by plasmid DNA isolation and sanger sequencing. The primers used were caaaccaggattctagcccata and gtggttgccaccttgttacc for _KRAS4A_ and ttcagttgcctgaaga-gaaaca and agtctgcatggagcaggaaa for _KRAS4B_. Human _RBM39_ and _CA9_ knock-down cells were generated by CRISPRi/dCas9-KRAB-mediated genome engineering as previously described[15]. A549 cells stably expressing dCas9-KRAB were enriched by flow cytometry for BFP expression and two sgRNA targeting _RBM39_ or _CA9_ were transfected into A549-dCas9-KRAB cell for puromycin selection. sgRNA targets were GAGCAGCGGCCGCCATTCA and GGA-GAGCAGGACGGCGGGCTT for _RBM39_ and GGGATCAACAGAGGGAGCCA and GCAGGGGCCGGGATCAACAG for _CA9_.

The shRNA-mediated knockdown cells were generated by antibiotic selection after infection with retroviral particles collected in supernatants of Phoenix cells transfected with pSuper.retro plasmid carrying the shRNA sequences. shRNA targets GGTGAGGGAGATCCGACAATA for _KRAS4A_ and GACAGGGTGTTGATGATGCCT for _KRAS4B_.

**Western Blotting analysis**. Cells were lysed with RIPA buffer (Thermo Scientific) and lysates concentrations were determined by BCA protein assay (Thermo Scientific). 80ug lysates were subjected to 4–12% SDS-PAGE (Bio-Rad) and then transferred to PVDF membrane. PVDF membranes were blocked in Tris-buffered saline 0.1% Tween-20 (TBST) containing 5% non-fat milk for 1 h at room temperature and incubated overnight with primary antibody diluted in TBST containing 3% non-fat milk at 4 °C. Membranes were washed with TBST and incubated with horseradish peroxidase (HRP)-conjugated secondary antibodies diluted in TBST containing 3% non-fat milk at 4 °C for 2 h. The primary antibodies used were Hras (sc520; Santa Cruz, 1:1000 dilution), Nras (sc519; Santa Cruz, 1:1000 dilution), phospho-p44/p42 Map Kinase (cat #2338, Cell Signaling Technology, 1:2000 dilution), phospho-Akt (cat #4060, Cell Signaling Technology, 1:2000 dilution), phospho-Erk (cat #4370, Cell Signaling Technology, 1:2000 dilution), β-actin (sc47778; Santa Cruz, 1:5000 dilution); KRAS4B (WH0003845M1; sigma, 1:2000 dilution); rat anti-KRAS4A (10C11E4, custom antibody, 1:100 dilution) and RBM39 (WH0009584M1; sigma, 1:2000 dilution). The secondary antibodies used were anti-mouse IgG-HRP (cat #7076, Cell Signaling Technology, 1:10000 dilution), anti-rabbit IgG-HRP (cat#7074,Cell Signaling Technology, 1:10000 dilution), and anti-Rat IgG-HRP (NA935, Millipore-Sigma, 1:2000 dilution). Proteins were visualized using the enhanced chemiluminescence (ECL) system (ECL™ Prime Western Blotting System, GE Healthcare Bioscience).

**Antibody generation**. Custom rat anti-KRAS4A antibody was developed by Genscript using the peptide sequence CEIRQYRLKKISKEEK as antigen for immunization.

**Cell proliferation assay**. A total 1,000 cells were plated in 96-well culture plates and cell proliferation was determined by CyQUANT cell proliferation assay kit (C35011, Invitrogen) as described by the manufacturer.

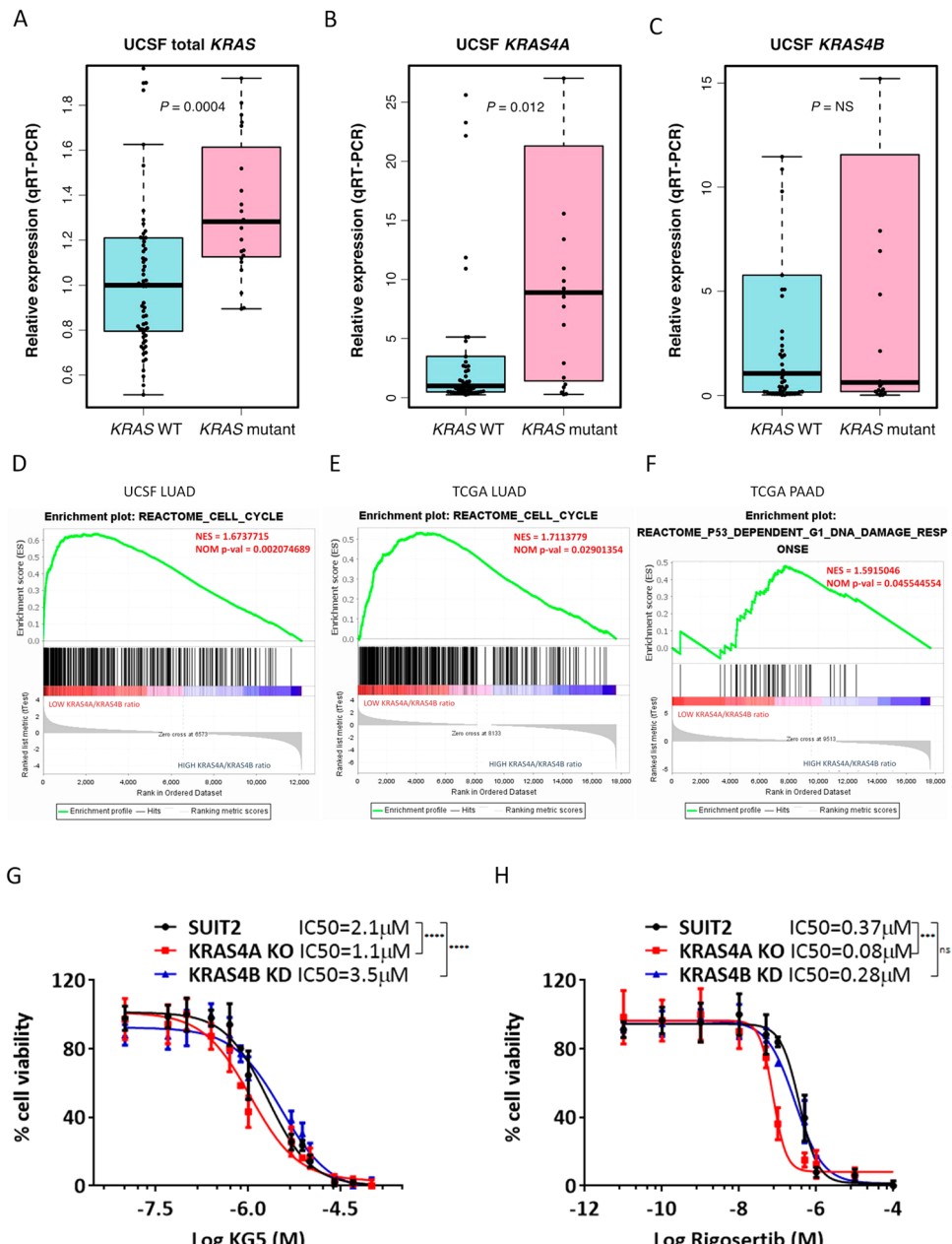

**Fig. 6 Low levels of *KRAS4A* are associated with cell cycle/DNA damage gene expression signatures. A** Total *KRAS*; **B** *KRAS4A*; and **C** *KRAS4B* expression in human lung cancers were determined by isoform-specific TaqMan assays. The number of individual lung cancer patients with wildtype *KRAS* tumors ($n = 63$) and mutant *KRAS* tumors ($n = 23$) were analyzed. The center line is the median, the bottom of the box is the 25th percentile boundary, the top of the box the 75th, and the whiskers define the bounds of the data that are not considered outliers, with outliers defined as greater/lesser than ± 1.5 × IQR, where IQR = inter quartile range. (**D–F**) GSEA plots for functional gene sets enriched in low *KRAS4A/KRAS4B* ratio tumors in UCSF (**D**) and TCGA lung (**E**) and pancreas (**F**) data sets. Tumors with a low ratio of *KRAS4A/KRAS4B* have higher expression of genes linked to cell cycle and mitosis. Treatment with KG5 (**G**) or Rigosertib (**H**) significantly impaired growth of the *KRAS4A* knockout SUIT2 cells compared to the parental or *KRAS4B* knockdown cells. Cells were treated with inhibitors for 72 h and cell counts measured using the Cyquant assay. Data are presented as mean ± s.d from $n = 7$ independent experiments in (**G**) and $n = 3$ independent experiments in (**H**). ***$P < 0.001$; ****$P < 0.0001$ by two-way ANOVA with Dunnett's multiple comparison.

**Colony formation**. Cells were plated at density of 500 cells per well in six-well plate and incubated for 10 days at 37 °C in DMEM medium containing 10% fetal bovine serum and 1% penicilin-streptamycin. Colonies were counted after methanol fixation and crystal violet staining.

**Soft agar growth assays**. In total 5,000 cells were mixed with 1 ml of 0.3% agar in DMEM supplemented with 10% FBS and layered onto 1.5 ml 0.6% agar in DMEM supplemented with 10% FBS. The colonies were incubated for 2 weeks in SUIT2 cells and for 3 weeks in A549 cells. At the end of incubation, the colonies were

stained with crystal violet and the images were captured using microscope followed by quantification of colony area using Image-Pro Plus software.

**Xenograft tumor model**. SUIT2 cells ($4 \times 10^6$) and A549 cells ($3 \times 10^6$) were dispersed in 75 μL DMEM and 75 μL Matrigel (356230, BD Biosciences) and injected subcutaneously into nude mice. Tumor volume was calculated as follows: $V = L \times W^2 \times 0.52$, where L and W represent length and width respectively. For the in vivo efficacy of indisulam study, NSG mice were implanted with $2 \times 10^6$ SUIT2 or $3 \times 10^6$ A549 mixed with matrigel. After cell implantation for 5 days, the

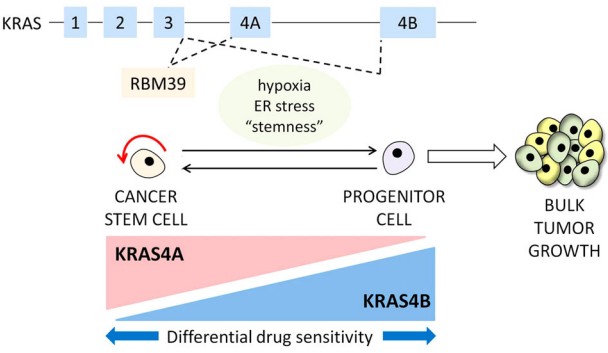

**Fig. 7 Model for control of stem-progenitor cell transition by KRAS4A/4B splicing.** KRAS4A is enriched in the cancer stem cell population, whereas KRAS4B is more ubiquitously expressed in stem and progenitor cells. Regulation of the expression of *KRAS4A*, in part through the DCAF15/ RBM39 splicing complex, helps to modulate the stress responses associated with the cancer stem-progenitor cell transition. Cells with a low KRAS4A/KRAS4B ratio show higher sensitivity to a subset of cancer therapeutics.

indisulam or vehicle was delivered for 8 days in SUIT2 and 5 days for A549 by retro-orbital injection.

**Flow cytometry**. For side population identification and isolation analysis, cells were suspended at $1 \times 10^6$ cells per ml in DMEM containing 2% FBS, 10 mM HEPES and 5 μg/ml Hoechst33342 dye (B2261, Sigma Aldrich), either alone or combination with 50 μM verapamil (ALX-550-306-G001, Enzo Life Sciences) and incubated in water bath at 37 °C for 2 h. After incubation, cells were suspended in PBS containing 2% FBS and 10 mM HEPES on ice and stained with 1 μg/ml propidium iodide followed by sorting or analysis using AriaII (BD) fluorescence activated cell sorting system (FACS). For the ALDH activity analysis, cells were suspended at $1 \times 10^6$ cells per ml in ALDEFLUOR assay buffer containing activated ALDEFLUOR™ Reagent, either alone or combination with diethyl amino-benzaldehyde (DEAB), and incubated 30 min at 37 °C according to the manu-facturer's instruction in ALDEFLUOR kit (01700, StemCell Technologies).

**RNA extraction and quantitative polymerase chain reaction (qPCR)**. RNA was extracted using TRIzol Reagent (15596026, Invitrogen), and cDNA was synthesized using iScript Synthesis kit (1708840, Bio-Rad) according to the manufacturer's instructions. Quantitative real-time RT-PCR was carried out using Taqman Mix in an ABI Prism7900HT Sequence Detection System (Applied Biosystems, Foster City, CA). TaqMan Gene expression assays used were as follows: *KRAS* (Hs00364282_m1), *KRAS4A* (Hs00932330_m1 KRAS), *KRAS4B* (Hs00270666_m1 KRAS), *HPRT1* (Hs02800695_m1) and *ABCG2* (Hs01053790_m1). The primers and probe used for amplification of *KRAS4A* were as follows: TGTGATTTGCCTTCTAGAACAGTAGAC, CTCACCAATGTA-TAAAAAGCATCCTC, and 5'-FAM- CAAAACAGGCTCAGGAC-MGB-3'. Relative mRNA expression levels were normalized to *HPRT1*.

**TCGA analysis of *KRAS4A* and *KRAS4B* isoform-specific expression**. TCGA lung adenocarcinoma (LUAD), pancreatic adenocarcinoma (PAAD), and col-orectal adenocarcinoma (COADREAD) RNA sequencing and clinical annotation data were downloaded from UCSC Cancer Genome Browser (now UCSC Xena). Specifically, level 3 normalized RSEM values for reads spanning splice junctions was downloaded, and used to calculate frequencies of reads spanning junctions of exons 4A and 4B of *KRAS*. Given that splicing to yield *KRAS4B* results in exclusion of exon 4A, while *KRAS4A* results in inclusion of both exons 4A and 4B, an exon 4A inclusion score was calculated to determine the fraction of *KRAS* transcripts made up by *KRAS4A*. This score is calculated as $((inc1 + inc2)/2)/ (((inc1 + inc2)/2) + exc)$, where inc1 and inc2 are reads spanning the junction of exons 3 and 4A, and reads spanning the junction of exons 4A and 4B, respec-tively, and exc are reads spanning the junction of exons 3 and 4B. To prevent artificially low scores in samples with lower overall *KRAS* expression or read coverage, particularly those that are *KRAS* WT, samples with less than 20 exc reads were not included in the analysis. No limit was set on inc1 and inc2 reads, as it is biologically plausible that some samples have minimal *KRAS4A* expression.

**Gene set enrichment analysis (GSEA)**. Microarray gene expression profiling of UCSF lung cancer tumors and TCGA RNAeq data were used for gene set

enrichment analyses. GSEA was performed using curated gene sets (c2.cp.reac-tome.v6.2.symbol.bmt).

**Data analysis and generation of plots**. Data were analyzed and non-parametric statistical tests performed in R. Plots were generated using the R package ggplot2 (H. Wickham. ggplot2: Elegant Graphics for Data Analysis. Springer-Verlag New York, 2009.)

**Reporting summary**. Further information on research design is available in the Nature Research Reporting Summary linked to this article.

## Data availability

The TCGA, CCLE, depmap and microarray data are publicly available datasets. The gene expression microarray data was downloaded from the NCBI Gene Expression Omnibus with accession numbers GSE33734 (https://www.ncbi.nlm.nih.gov/geo/query/acc.cgi? acc=GSE33734). The gene expression data from the TCGA project were downloaded from the UCSC cancer browser (http://xena.ucsc.edu/welcome-to-ucsc-xena/). The drug sensitivity was downloaded from depmap portal (https://depmap.org/portal/). The gene expression was downloaded from CCLE (https://portals.broadinstitute.org/ccle). All relevant data are available from the authors. Source data are provided with this paper. The remaining data are available within the Article, Supplementary Information or available from the authors upon request. Source data are provided with this paper.

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

## Acknowledgements

We are greatly appreciative of the help and comments from our colleagues in refining this study and manuscript. This work was supported by US National Cancer Institute (NCI) grants RO1CA184510, UO1 CA176287, and R35CA210018 and the Barbara Bass Bakar Professorship of Cancer Genetics. Wei-Ching Chen was supported by a Fellowship from Taiwan Ministry of Science and Technology and by the UCSF Pancreas Center and the Schwartz Family Foundation. P.M.K.W was supported by NIH training grant T32 GM007175 and an NCI F31 NRSA award. M.D.T. was supported by a UCSF Senate Research grant. S.R.B was supported by NIGMS Predoctoral Training in Biomedical Sciences T32 GM008568 (Ansel).

## Author contributions

W.C.C. designed and carried out most of the biochemical experiments shown in Figs. 2–6 for characterization of *Kras4A* and *Kras4B* cells, and generated Crispri/dCas9-KRAB knockdown cells, M.D.T. helped with the overall study design, generated the *Kras4B* KO mouse, carried out shRNA studies and mouse carcinogenesis experiments, P.W. generated and characterized Crispr/Cas9 knockout cells and carried out bioinformatics analysis, animal carcinogenesis studies and tumor analysis, M.P. provided guidance and protocols for biochemical and metabolic assays and helped with data analysis and interpretation, R.D. performed mouse carcinogenesis studies, I.J.K. carried out TaqMan analysis of *KRAS4A/B* expression in human tumours, Q.T. performed mouse tumour mutation analysis and helped with cell culture experiments, N.B. characterized Crispr/Cas9 knockout cells, S.R.B. carried out bioinformatics analysis, H.G. carried out analysis of RBM39 binding sites, O.M. carried out inhibitor studies with human tumor cell lines, A.B. designed and supervised the overall study. The paper was written by A.B., M.D.T., W.C.C., and P.W., with contributions from the other authors.

## Competing interests

The authors declare no competing interests.
