## [Peer Review File · Nature Communications]

REVIEWER COMMENTS

Reviewer #2 (Remarks to the Author): Expert in mouse lung cancer models

Chen et al. investigated the role of the KRAS4A and KRAS4B splice variants on tumor initiation in vivo and in cell lines. They claim that it is necessary for both variants to be expressed from the same allele in order to induce lung tumors in a chemical assay. They show that the KRAS4A variant is enriched in the side population, and that cells upregulate the two variants depending on the type of stress they are exposed to. Furthermore, they show that inhibition of RBM39/DCAF15 downregulates KRAS4A and decreases tumor formation in transplantation assays. The authors claim that the observed reduction is due to the specific targeting of a cancer stem cell population that has higher levels of KRAS4A.

There are multiple issues with the way data was interpreted, and more controls are necessary to address these concerns.

1. There were no tumors in the *Kras4A*^{-/-} or *Kras4B*^{-/-} homozygous mice, or in the double heterozygous animals even after repeated and high doses of urethane. In the single heterozygous mice, all mutations were in the remaining wildtype allele. Has it ever been established that urethane can in fact cause mutations in these cDNA versions of the gene? Maybe it is necessary to have the full gene, including introns, to get mutations in a normal frequency using urethane. That needs to be tested, otherwise the results cannot be interpreted in the way the authors did.
2. The authors use side population cells to make the claim that KRAS4A is enriched in cells with cancer stem cell properties. However, they do not show any FACS data to show how they obtained side population cells, and there is no functional validation that this population is actually enriched in cancer stem cells in the cell lines they used. The authors need to show that sorted side population cells are better in growing tumors than the rest of the cells. Showing levels of aldehyde dehydrogenase is not sufficient to make that claim.
3. The authors treated mice that were transplanted with cell lines with indisulam and observed reduced in vivo tumor growth. However, as a “general inhibitor of the splice machinery” that “influences splicing of many pre-mRNAs genome-wide”, the treatment with indisulam probably has broad and unpredictable effects. A rescue assay is therefore necessary to show that specifically reduction of KRAS4A is responsible for the observed growth reduction, e.g. by overexpressing KRAS4A in the cell lines at the same time.
4. In Figure 2A, SUIT2 KRAS4B KD also has reduced levels of KRAS4A, and A549 KRAS4A KO has reduced levels of KRAS4B, questioning the specificity of the used sgRNAs. Furthermore, KRAS levels are overall reduced because of the knockouts. Could a general reduction of KRAS be enough to cause the observed changes? This should be tested, e.g. by overexpressing KRAS4A in the 4B knockouts and vice versa.

Minor comments:

1. The double heterozygous mice should not be referred to as *Kras4A*^{+/-} ; *Kras4B*^{+/-} , as this implies

that there is still a functional wildtype allele. Kras4A /4B would be more appropriate.

2. The authors show that “Kras4A and Kras4B are both activated in response to EGF in cells that lack the alternative isoform (Extended Data Fig. 2)”. However, activation of MapK and Akt pathways could also be mediated through other Ras proteins.

3. The authors find enrichment of cell cycle genes in the low KRAS4A/KRAS4B ratio patient tumor samples. However, growth is impaired when they knock out KRAS4A in their cell line assays. How do the authors explain this discrepancy?

4. Figure 6G+H no IC50 value is stated, no statistical test or p-value is indicated.

Reviewer #3 (Remarks to the Author): Expert in mouse lung cancer models

Chen and colleagues investigate the roles of KRAS4A and 4B isoforms in development of KRAS mutant tumors in this manuscript. Their major findings are: (a) coordinated regulation of both isoforms is essential for development of KRAS-mutated tumors, (b) less common KRAS4A isoform is enriched in cancer stem-like cells, with its splicing controlled by DCAF15/RBM39 pathway, and (c) KRAS4A/4B ratio may be a promising biomarker for therapeutic prediction. Their conclusions are further strengthened by evaluation of their findings in the UCSF lung adenocarcinoma patient samples.

Major comments/suggestions:

1) Fig 2 – Since oncogenic KRAS mutations are heterozygous, what is the impact of KRAS 4A or KRAS 4B KD on the allelic ratio of the mutant to WT allele? Is it maintained or is the oncogenic allele selectively depleted?

2) Fig 3 - Use of indisulam is complicated by the fact that it is also a CDK inhibitor. Does over-expression of an oncogenic KRAS4A allele rescue its effects in SU12758 and A549 cells? How does this compare to over-expression of a KRAS4B allele?

3) In Figs. 3I and J, the authors conclude that the tumors with the lower ratio of KRAS4A/4B are more susceptible to indisulam, yet based upon their model one would have expected the opposite result. Can the authors include lung and pancreatic cancer cell lines in Fig. 3J (DEPMAP analysis of sensitivity to indisulam) as well? Although lung cancers are heterogeneous in terms of KRAS mutations, 90% of pancreatic tumors have KRAS mutations. Does indisulam impact KRAS splicing in representative Ewing's sarcoma or leukemia/lymphoma lines? Perhaps this result is due to a different mechanism of action (eg CDK inhibition) in this context. Furthermore, what is the effect of indisulam and RBM39 knockout on KRAS WT human lung cancer cell lines?

4) Fig 4 - while stem cell “side population” cells are approximately 1%, ALDH-positive cells are approximately 15% in Fig. 4C and 4D. It would be helpful to simultaneously stain for ALDH and other potential markers of cancer stem cell-like cells such as CD166 to identify double positive cells, and repeat experiments in Fig. 4D to improve the robustness of these findings.

5) Fig 6 – several additional interpretations of these results should be considered. Again this supports a potential relationship between the CDK inhibitor effects of indisulam and preferential sensitivity of the low ratio KRAS4A/B cancers which are more proliferative. More interestingly have the authors looked at cell cycling in the stem-like cells that preferentially express KRAS4A? Perhaps this supports a more quiescent like state that has been linked to stem cell behavior.

Minor points:

- Typo in Fig. 6 legend: “liung’ instead of ‘lung’

Reviewer #4 (Remarks to the Author): Expert in splicing and cancer

Review of “Targeting KRAS4A splicing through the RBM39/DCAF15 pathway inhibits cancer stem cells” by Chen et al.

In this manuscript, Chen et al. propose that the minor splicing isoform of KRAS, KRAS4A, is a key cancer gene in tumor stem cells. KRAS4A splicing (exon inclusion/skipping) is regulated by RBM39 and the molecular glue sulfonamides that degrade RBM39. Based on these mechanisms, the authors suggest to a biomarker strategy for clinical application of these RBM39/DCAF15-targeting molecules. Overall, the experiments were well done but I am afraid that the mechanisms and proposed translational applications are not fully justified. In particular, it is not convincing that the antitumor activity by KRAS4A inhibition is potent enough, even under the KO condition (Fig. 1E). The definition of “stem cell” or side-population is questionable by solely using the gene markers (Fig. 4). In addition, as RBM39 is an essential gene (in DepMap) which regulates a relatively large set of RNA splicing, the authors need to show direct evidence supporting that the antitumor activity by indisulam is through inhibition of KRAS4A splicing (Fig. 3H).

Specific points:

1. Fig. 3A~D, the concentrations of splicing modulators used here were very high (5~20 microM) and time course of treatment was long (48h). Can authors be certain of the fitness of the cells, which may impact the splicing alterations as a secondary event?
2. KRAS4A cDNA rescue in Fig 3H is crucial to demonstrate the antitumor activity of indisulam is indeed associated with KRAS4 splicing in these KRAS-mutant cancer cell lines.
3. It would be more convincing to combine the CCLE/DepMap data shown in Fig. 3I and J: KRAS4A/4B ratio vs. indisulam AUC at cell line level. A positive correlation is anticipated to suggest the relatively low level of 4A offers sensitivity to RBM39 degradation. How do we explain the association?
4. The data shown in Fig. 6G, particularly for KG5, is not convincing. There was basically only one concentration point showing difference. More dilutions might help.

REVIEWER COMMENTS

Responses in blue italics

Reviewer #2 (Remarks to the Author): Expert in mouse lung cancer models

Chen et al. investigated the role of the KRAS4A and KRAS4B splice variants on tumor initiation in vivo and in cell lines. They claim that it is necessary for both variants to be expressed from the same allele in order to induce lung tumors in a chemical assay. They show that the KRAS4A variant is enriched in the side population, and that cells upregulate the two variants depending on the type of stress they are exposed to. Furthermore, they show that inhibition of RBM39/DCAF15 downregulates KRAS4A and decreases tumor formation in transplantation assays. The authors claim that the observed reduction is due to the specific targeting of a cancer stem cell population that has higher levels of KRAS4A.

There are multiple issues with the way data was interpreted, and more controls are necessary to address these concerns.

1. There were no tumors in the *Kras4A*^{-/-} or *Kras4B*^{-/-} homozygous mice, or in the double heterozygous animals even after repeated and high doses of urethane. In the single heterozygous mice, all mutations were in the remaining wildtype allele. Has it ever been established that urethane can in fact cause mutations in these cDNA versions of the gene? Maybe it is necessary to have the full gene, including introns, to get mutations in a normal frequency using urethane. That needs to be tested, otherwise the results cannot be interpreted in the way the authors did.

Response: We agree that it is important to demonstrate that these knock-in cDNA constructs in the Kras locus can in fact be mutated by carcinogens. This possibility has however been tested in a previous study in which mice carrying exactly the same kind of knock-in cDNA construct expressing Hras under the control of the Kras gene promoter were treated with carcinogens, leading to development of lung cancers in which the cDNA construct carried the appropriate carcinogen-specific mutation (To et al, Nat Genet. 2008 Oct; 40(10): 1240–1244). All knock-in constructs expressing Kras4A, Kras4B, or Hras have the same conserved Kras 5' sequence and differ only at the 3' variable region responsible for membrane localization of the Ras protein. This Hras knock-in cDNA mouse has also been used for other carcinogenesis studies in the skin, in which it was shown that carcinogen treatment results in somatic mutations that are dependent on the nature of the carcinogen used. We conclude that introns are not necessary for mutations, and that the engineered cDNA constructs are very good targets for mutagenesis. The To et al reference was cited in our original submission (Ref 6), but because of space limitations we did not include any detailed comparison with the present study using Kras4A or Kras4B constructs. In the revised version of the manuscript, we have now included a brief discussion of this interesting point in the Results section.

2. The authors use side population cells to make the claim that KRAS4A is enriched in cells with cancer stem cell properties. However, they do not show any FACS data to show how they obtained side population cells, and there is no functional validation that this population is actually enriched in cancer stem cells in the cell lines they used. The authors need to show that sorted side population cells are better in growing tumors than the rest of the cells. Showing levels of aldehyde dehydrogenase is not sufficient to make that claim.

Response: The isolation of “side population” cells by a standard protocol, followed by the demonstration of enrichment for stem cell properties such as growth as spheroids has been carried out many times and several publications describing these procedures were cited in our original manuscript (Orecchioni et al, Methods Mol Biol. 2016; 1464: 49-62) (Ho et al, Cancer Res. 2007 May; 67(10): 4827-4833) (Hirschmann-Jax et al, Proc Natl Acad Sci U S A. 2004 Sep; 101(39): 14228-33). In response to the reviewer’s comments, we have repeated these studies to isolate side population cells and analyze sphere formation efficiency of side population and non-side population cells from 3 different cell lines: SUIT2, A549 and AsPC1. The gating strategy for side population cell isolation is shown in Fig R1A, and Fig R1B shows that sorted side population cells show significantly better clonal growth in ultra-low attachment plate than the non-side population cells from the same cell line. These data are included in Extended data Fig 8 in the revised manuscript.

A

B

Fig. R1. (A) Gating strategy for side population cells. Side population cells were identified in SUIT2, A549 and AsPC1 following Hoechst 33342 staining in the presence or absence of verapamil by FACS analysis using blue and red-wavelength assessment. (B) Graph quantifying the sphere formation efficiency in ultra-low attachment plate of side population cells derived from SUIT2, A549 and AsPC1 cell lines. Representative images of spheres derived from AsPC1 cells.

3. The authors treated mice that were transplanted with cell lines with indisulam and observed reduced in vivo tumor growth. However, as a “general inhibitor of the splice machinery” that “influences splicing of many pre-mRNAs genome-wide”, the treatment with indisulam probably has broad and unpredictable effects. A rescue assay is therefore necessary to show that specifically reduction of KRAS4A is responsible for the observed growth reduction, e.g. by overexpressing KRAS4A in the cell lines at the same time.

This is an interesting point, also raised by other reviewers, about the effects of indisulam on cell growth, suggesting KRAS4A rescue experiments.

We did in fact carry out growth rescue experiments during the early stages of this work, and they did not give clear cut results, we believe for the following reasons:

1. Rescue experiments are optimal when replacement of the deleted target restores its immediate downstream function. The effect of indisulam on KRAS4A levels is indirect, first by binding to Dcaf15, then tethering of RBM39 leading to its degradation, followed by loss of splice control of KRAS4A, leading to reduction in the proportion of stem cells in human cancer cell lines (see Figure). The effect of over-expressing KRAS4A on the cellular growth response to indisulam is very complex, as loss of KRAS4A is several steps removed from the initial effect of indisulam, each of which has many off target effects that complicate the interpretation of any results obtained. Indisulam is a non-specific inhibitor of many other targets including CDKs, as mentioned by the reviewer, as well as carbonic anhydrase. It binds to Dcaf15, a component of the Cul4-RING E3 ubiquitin ligase complex which has hundreds of potential binding partners, resulting in degradation of RBM39, which also has many splicing targets genome-wide. We believe that this is one reason for the lack of consistent results when we tried to do the growth rescue experiment. We did however demonstrate that loss of cancer stem cells by reduction in KRAS4A levels was rescued by direct over-expression of exogenous KRAS4A as shown in the original manuscript Fig 4.

2. The growth rescue experiment is also complicated by the fact that KRAS4A is expressed primarily in the stem cell-like cells in tumor cell lines, but transfection followed by selection will introduce the construct into **all** cells that survive selection. This may also perturb the cell population dynamics of these cells in culture, again making the interpretation of the growth patterns more difficult. This is further complicated by the myriad documented effects of over-expressing RAS in cells in culture, which can lead to growth arrest, growth stimulation, or increased differentiation. These effects are notoriously dose-dependent, suggesting that any rescue effect may require finding the correct concentration of KRAS4A in the correct (ie stem-like) cell population, which is obviously technically challenging (see also response to question 5 below).

We believe that the link between indisulam and KRAS4A, while not direct, is however supported by the additional experiments carried out in response to the comments of reviewer 2 (see below) and included in the revised manuscript. These studies, using a range of DEPMap cell lines with differing KRAS4A levels, show that indisulam treatment has the strongest effects in cells with the lowest KRAS4A levels, and suggest that an assay for KRAS4A levels may be used as a biomarker for the use of indisulam to treat cancer patients.

4. In Figure 2A, SUI2 KRAS4B KD also has reduced levels of KRAS4A, and A549 KRAS4A KO has reduced levels of KRAS4B, questioning the specificity of the used sgRNAs.

Response: We have repeated these experiments many times, with consistent results which are also supported by the sequencing of the CRISPR knockout clones used, attesting to the specificity of the guide RNAs. There is some inter-experimental variation in the western blots, and quantification of KRAS4A and KRAS4B levels using imageJ from four biological replicate experiments confirms the specificity of the sgRNAs used for the CRISPR knockouts (Fig R2A, B). A more representative Western blot is also shown in Fig 2A in the revised manuscript.

A

B

C

Fig. R2. (A) Graph quantifying the KRAS4A and KRAS4B levels in CRISPR-Cas9 induced knockouts of KRAS4A and KRAS4B in SUIT2 and A549 cells. The quantification results are based on the different Western Blots shown in Fig R2B. (C) Over-expression of doxycycline-inducible KRAS4A or KRAS4B can increase growth in culture of parental A549 and KRAS4A knockout cells. Parental A549 cells and KRAS4A knockout cells were transfected with Dox-inducible KRAS4A or KRAS4B, and growth in 2D culture was assessed after treatment with doxycycline or control medium.

5. Furthermore, KRAS levels are overall reduced because of the knockouts. Could a general reduction of KRAS be enough to cause the observed changes? This should be tested, e.g. by overexpressing KRAS4A in the 4B knockouts and vice versa.

Response: It is certainly possible that total KRAS levels may affect the growth of cells in vitro, regardless of isoform specificity. It is known that the “Rasless MEFS” generated by the Barbacid group in which all three Ras genes have been deleted (Drosten et al, EMBO Journal (2010) 29, 1091–1104) can be rescued (in terms of growth) by expression of different Ras isoforms.

We carried out growth restoration assays using the CRISPR knockout cells and found, as shown in Fig R2C, that expression of FLAG-tagged KRAS4A or KRAS4B can increase the growth of parental A549 cells, and also induced a modest increase in growth of KRAS4A knockout cells. Results obtained using the KRAS4B knockdown cells were inconsistent, possibly due to the fact that these cells only had partial loss of KRAS4B expression.

We conclude, as also demonstrated by others, that total RAS can affect the growth properties of cells in culture, but this does not say anything about the specific functions of different Ras isoforms in distinct cells within these bulk cultures.

Minor comments:

1. The double heterozygous mice should not be referred to as *Kras4A*^{+/-} ; *Kras4B*^{+/-} , as this implies that there is still a functional wildtype allele. *Kras4A/4B* would be more appropriate.

Response: Thank you for this comment. The nomenclature used for the double heterozygotes has been changed in the revised version of the manuscript.

2. The authors show that “*Kras4A* and *Kras4B* are both activated in response to EGF in cells that lack the alternative isoform (Extended Data Fig. 2)”. However, activation of MapK and Akt pathways could also be mediated through other Ras proteins.

Response: We agree, but showed these data to demonstrate that the canonical Ras signaling pathways were not dramatically affected by loss of either KRAS isoform.

3. The authors find enrichment of cell cycle genes in the low KRAS4A/KRAS4B ratio patient tumor samples. However, growth is impaired when they knock out KRAS4A in their cell line assays. How do the authors explain this discrepancy?

*Response: This is a very interesting point. In human cell lines, we consistently find that downregulation of KRAS4A by CRISPR/Cas9 knockout or using shRNA leads to suppression of growth after transplantation into immunodeficient mice. Because the TCGA data are based on human patient samples, we carried out a preliminary study to ask whether the immune system can affect the growth of mouse cells in which *Kras4A* has been suppressed using shRNA. These preliminary experiments suggested that the immune system does affect the growth of cells with low levels of *Kras4A*, and that this could be an explanation for the difference between*

the in vitro results and data from TCGA. However these experiments, which are presently being repeated, raise many additional questions that we believe are outside the scope of the present manuscript. In the revised manuscript, we have mentioned the discrepancy pointed out by the reviewer between patient tumor samples and the cell lines, suggested possible immune system involvement, and stated that these questions will be addressed in future studies.

4. Figure 6G+H no IC50 value is stated, no statistical test or p-value is indicated

Response: Thank you for pointing out this error. The revised manuscript now includes the information on IC50 levels and p values.

Reviewer #3 (Remarks to the Author): Expert in mouse lung cancer models

Chen and colleagues investigate the roles of KRAS4A and 4B isoforms in development of KRAS mutant tumors in this manuscript. Their major findings are: (a) coordinated regulation of both isoforms is essential for development of KRAS-mutated tumors, (b) less common KRAS4A isoform is enriched in cancer stem-like cells, with its splicing controlled by DCAF15/RBM39 pathway, and (c) KRAS4A/4B ratio may be a promising biomarker for therapeutic prediction. Their conclusions are further strengthened by evaluation of their findings in the UCSF lung adenocarcinoma patient samples.

Major comments/suggestions:

1) Fig 2 – Since oncogenic KRAS mutations are heterozygous, what is the impact of KRAS 4A or KRAS 4B KD on the allelic ratio of the mutant to WT allele? Is it maintained or is the oncogenic allele selectively depleted?

Response: It is certainly true that many human and mouse tumors have heterozygous KRAS mutations, and in these the balance between expression of the wild type and mutant KRAS alleles can affect tumor growth and other properties such as metabolism (Kerr et al, Nature volume 531, pages110–113(2016)). In the human cells used in this study, all were homozygous for their respective RAS mutations, so the question of the effect of deletion of one isoform on copy number of the oncogenic allele does not apply.

2) Fig 3 - Use of indisulam is complicated by the fact that it is also a CDK inhibitor. Does over-expression of an oncogenic KRAS4A allele rescue its effects in SUIT2 and A549 cells? How does this compare to over-expression of a KRAS4B allele?

Response: Thank you for raising this point, which was also mentioned by the other reviewer. Please see the responses above. Indisulam controls KRAS4A indirectly through degradation of RBM39, and also has many other targets that can affect tumor growth. For these and other reasons, obtaining consistent and interpretable results using these rescue approaches was not possible.

3) In Figs. 3I and J, the authors conclude that the tumors with the lower ratio of KRAS4A/4B are more susceptible to indisulam, yet based upon their model one would have expected the opposite result.

Response: We envisage 2 possible explanations of the increased susceptibility of tumors with low KRAS4A levels to indisulam treatment (which was confirmed and extended as described below). The first is that if KRAS4A levels are already low, treatment with indisulam can reduce the level even further, taking it below a critical threshold level for cell growth. In support of this interpretation, we have now shown that while KRAS4A and RBM39 are downregulated significantly by low doses of indisulam in HL60 and Jurkat cells, the same dose when applied to SUIT2 cells, which have high KRAS4A levels, has only minimal impact on KRAS4A, and reduced potency as a growth inhibitor (Fig R3E).

An alternative (or additional) explanation is that low levels of KRAS4A are associated with higher levels of G2/M checkpoint genes, making these cells more vulnerable to the activity of indisulam as a CDK inhibitor. These explanations are not mutually exclusive, and presently we can not distinguish between them without carrying out extensive additional mechanistic studies. Our new data do however support the possibility proposed in the original manuscript that levels of KRAS4A can be a biomarker for sensitivity to indisulam, and possibly to other cancer drugs.

Can the authors include lung and pancreatic cancer cell lines in Fig. 3J (DEPMAP analysis of sensitivity to indisulam) as well? Although lung cancers are heterogeneous in terms of KRAS mutations, 90% of pancreatic tumors have KRAS mutations. Does indisulam impact KRAS splicing in representative Ewing's sarcoma or leukemia/lymphoma lines? Perhaps this result is due to a different mechanism of action (eg CDK inhibition) in this context. Furthermore, what is the effect of indisulam and RBM39 knockout on KRAS WT human lung cancer cell lines?

We have obtained several of the cell lines characterized in DEPMAP and have repeated cell growth assays in the presence of indisulam, and have also directly measured the levels of expression of the KRAS isoform to definitively establish whether there is a significant effect of KRAS4A on indisulam sensitivity. The sensitivity of indisulam across blood, lung and pancreas cancer cell lines is shown in Fig R3A. We chose 2 blood cancer cell lines (Jurkat and HL60), 2 lung cancer cell lines (NCI-H661 and NCI-H1703), and 2 pancreatic cancer cell lines (YAPC and SW1990) for these studies. Fig R3B shows that blood cancer cells are more sensitive to indisulam than lung NCI-H661 and pancreatic cancer cell lines, in agreement with the results from DEPMAP analysis. We have also determined the KRAS4A and KRAS4B levels in these 6 cell lines by western blot. YAPC shows the highest KRAS4A level and HL60 has lower KRAS4A/KRAS4B ratio than lung and pancreatic cancer cells (Fig R3C), in agreement with the results from CCLE data. Fig R3D shows that treatment with indisulam downregulated RBM39 and KRAS4A, but not KRAS4B, in Jurkat and HL60 cells. Compared to the effect of indisulam in SUIT2, downregulation of KRAS4A by indisulam was seen at lower concentration in HL60 (Fig R3E). Based on the positive correlation of lower KRAS4A/4B ratio with indisulam sensitivity, we would anticipate that lower KRAS4A/4B ratio can be a biomarker of sensitivity.

We also looked at the sensitivity of tumor cells with WT KRAS to treatment with Indisulam, and demonstrate that in H1650 cells which have a WT KRAS gene, KRAS4A is downregulated in response to both indisulam and knockdown of RBM39 (Fig R3F).

A indisulam (CTRP:411874) Drug sensitivity AUC (CTD²)

B

C loading lysate : 5μg

D

E loading lysate: 30μg

loading lysate: 30μg

F

Fig. R3. (A) DEPMAP analysis of sensitivity to indisulam for blood, lung and pancreas cancer cell lines. (B) The effects of indisulam on growth of a range of cancer cell types from blood, lung and pancreas, showing the strongest inhibition in HL60 and Jurkat cells. (C) Western blot of KRAS4A and KRAS4B for blood, lung and pancreas cancer cell lines. (D) The effect of indisulam on KRAS4A and RBM39 levels in leukemia cell lines. (E) The effects of indisulam on KRAS4A and KRAS4B levels in leukemia and pancreas cancer cell lines. Lower panels show the quantification using ImageJ of KRAS4A and KRAS4B levels after indisulam treatment. (F) The effect of indisulam on KRAS4A levels in WT KRAS and RBM39 knockdown human H1650 lung cancer cells.

4) Fig 4 - while stem cell “side population” cells are approximately 1%, ALDH-positive cells are approximately 15% in Fig. 4C and 4D. It would be helpful to simultaneously stain for ALDH and other potential markers of cancer stem cell-like cells such as CD166 to identify double positive cells, and repeat experiments in Fig. 4D to improve the robustness of these findings.

Response: We have combined CD166 and CD133 markers with ALDH for FACS analysis in CRISPR-Cas9 induced knockouts of KRAS4A and KRAS4B in SUIT2 and A549 cells. CD166 is expressed in most of the cells, and therefore does not seem to be a marker of a sub-population of stem-like cells in the SUIT2 and A549 tumor cell lines. The proportion of double positive ALDH+CD166+ cells shown in Fig R4A is similar to the single positive ALDH+ population shown in manuscript Fig 4D. CD133 is expressed in a subpopulation of SUIT2 cells, and the proportion of double positive ALDH+CD133+ cells in Fig R4B is lower than the single positive ALDH+ cells in Fig4D, suggesting that the combination of CD133 with ALDH improves characterization of cancer stem cell-like cells. This was only seen in SUIT2 cells, and CD133 did not provide any additional enrichment for isolation of stem cell populations from A549 cells. Figure R4A and B respectively show the results obtained using CD166 and CD133 for further enrichment of side population stem cells.

Fig. R4. Quantification of ALDH+CD166+ populations (A) and ALDH+CD133+ populations (B) in SUIT2 and A549 cells. CD166 did not provide any additional enrichment for side population stem cells, which was only seen for CD133 in SUIT2 cells.

5) Fig 6 – several additional interpretations of these results should be considered. Again this supports a potential relationship between the CDK inhibitor effects of indisulam and preferential sensitivity of the low ratio KRAS4A/B cancers which are more proliferative. More interestingly have the authors looked at cell cycling in the stem-like cells that preferentially express KRAS4A? Perhaps this supports a more quiescent like state that has been linked to stem cell behavior.

Response: This is a very good suggestion. Stem cells are mostly in a quiescent state compared to non-stem cells (Dean ST et al, Nat Rev Cancer. 2005 Apr; 5(4): 275-84). GF11 and Ncdin (NDN), as p53 target genes, have been identified as regulators of quiescence (Liu et al, Cell Stem Cell. 2009 Jan; 9:4(1):37-48) (Hock et al, Nature 2004 Oct; 431(7011):1002-7). DNA replication licensing factor MCM7 is lower in the side population cells and is induced from Go to G1 transition, suggesting that side population cells are mainly quiescent (Orr et al, Oncogene. 2010 Jul; 29(26):3803-14) (Ho et al, Cancer Res. 2007 May; 67(10): 4827-4833). The data in Fig R5 show that expression of MCM7, a marker of active cell cycle, was decreased while GF11 and NDN were increased, in side population cells from malignant pleural mesothelioma (GSE33734 dataset), in agreement with published data that side population cells are mostly quiescent. In response to the reviewer's comments, we have also investigated the KRAS4A levels from same GSE33734 dataset. The ILMN_1652104 probe is located within KRAS exon4A and the data in Fig R5 show that the KRAS4A level in side population cells is higher than in non-side population cells. ABCG2 expression was also enriched in side population cells, in agreement with our data in manuscript Fig 4A and Fig 4B. Taken together, these data provide strong support for the proposal that KRAS4A marks a stem-like cell population that is largely quiescent in human tumors.

Fig. R5. Expression of KRAS4A, ABCG2, MCM7, GF11 and NDN in side population and non-side population cells from the malignant pleural mesothelioma gene expression microarray GSE33734 dataset.

Minor points:

- Typo in Fig. 6 legend: "liung" instead of 'lung'

Corrected

Reviewer #4 (Remarks to the Author): Expert in splicing and cancer

Review of "Targeting KRAS4A splicing through the RBM39/DCAF15 pathway inhibits cancer stem cells" by Chen et al.

In this manuscript, Chen et al. propose that the minor splicing isoform of KRAS, KRAS4A, is a key cancer gene in tumor stem cells. KRAS4A splicing (exon inclusion/skipping) is regulated by RBM39 and the molecular glue sulfonamides that degrade RBM39. Based on these mechanisms, the authors suggest to a biomarker strategy for clinical application of these RBM39/DCAF15-targeting molecules. Overall, the experiments were well done but I am afraid that the mechanisms and proposed translational applications are not fully justified. In particular, it is not convincing that the antitumor activity by KRAS4A inhibition is potent enough, even under the KO condition (Fig. 1E). The definition of "stem cell" or side-population is questionable by solely using the gene markers (Fig. 4). In addition, as RBM39 is an essential gene (in DepMap) which regulates a relatively large set of RNA splicing, the authors need to show direct evidence supporting that the antitumor activity by indisulam is through inhibition of KRAS4A splicing (Fig. 3H).

Response: We thank the reviewer for the positive comments regarding the overall quality of the experiments.

The question about whether "the antitumor activity by KRAS4A inhibition is potent enough (Fig 1E)" was puzzling, as Fig.1E shows that Kras4A is absolutely required for tumor development in the mouse lung. No tumors developed in the homozygous Kras4A null mice even after repeated carcinogen treatment. A more modest effect on growth of human tumor cell lines is shown in Fig. 2, but this is what we would expect for inhibition of the minor KRAS4A isoform. Others have shown that even inhibition of total mutant KRAS, while causing transient growth inhibition, is incomplete and is followed by outgrowth of resistant cells. We are proposing that due to its expression in a sub-population of cancer stem-like cells, inhibition of KRAS4A will impact the dynamics of the cancer stem-progenitor cell transition and consequently potentiate the effects of other drugs.

Several of the other points raised were also made by other reviewers, and have been addressed in the responses above. Reviewer 2 also asked about the functional properties of side population stem cells (question 2). We therefore carried out extensive studies to isolate side population cells from 3 different cell lines and analyzed sphere formation efficiency (Fig. R1). Reviewer 3 requested studies using additional markers that have been reported to be enriched in side population cells from certain cell lines, and also asked about expression of markers of quiescence or cell cycle activation. The experiments carried out in response to these questions are presented in Fig. R4 and Fig. R5. Taken together, these studies demonstrate that side population cells are enriched in accepted stem-like properties such as clonal growth as spheroids (as frequently reported by others), but also that KRAS4A is enriched in human side population cells, and that these are in a more quiescent state.

Reviewer 2 also raised the point about the role of KRAS4A in growth inhibition induced by indisulam. As explained in some detail in the response to reviewer 2, this experiment, which we have carried out, did not give consistent results because of the fact that indisulam, as well as its target DCAF15 and degradation substrate RBM39, have many different downstream targets including CDKs that are known to be involved in growth control. We were able to restore the effects of KRAS4A inhibition on stem cells by over-expressing exogenous KRAS4A as shown in manuscript Fig 4, but effects on growth control were inconsistent for the reasons described above.

Specific points:

1. Fig. 3A~D, the concentrations of splicing modulators used here were very high (5~20 microM) and time course of treatment was long (48h). Can authors be certain of the fitness of the cells, which may impact the splicing alterations as a secondary event?

Response: These original concentrations were based on previous literature on the effects of indisulam on cell growth in vitro. In response to the reviewer's question, we have repeated the studies of the effect of indisulam at a range of concentrations on RBM39 levels and KRAS4A splicing. The data shown in Fig R6 demonstrate an effect of indisulam on RBM39 and KRAS4A levels in both SUIT2 and A549 cells, which is dose-dependent and seen at concentrations as low as 0.1 μ M.

Fig. R6. Dose-dependent effects of indisulam on RBM39 and KRAS4A levels.

2. KRAS4A cDNA rescue in Fig 3H is crucial to demonstrate the antitumor activity of indisulam is indeed associated with KRAS4 splicing in these KRAS-mutant cancer cell lines.

This is an important question that was discussed in detail above, and in the response to reviewer 2.

3. It would be more convincing to combine the CCLE/DepMap data shown in Fig. 3I and J: KRAS4A/4B ratio vs. indisulam AUC at cell line level. A positive correlation is anticipated to suggest the relatively low level of 4A offers sensitivity to RBM39 degradation. How do we explain the association?

In response to a similar question (3) from reviewer 3, we carried out extensive additional studies to investigate the relationship between KRAS4A levels and indisulam sensitivity. The data are now presented above in Fig. R3, and in the manuscript in Extended data Fig. 7. Thoughts on possible mechanisms that may explain the relationship between KRAS4A level and sensitivity to indisulam are also presented above, but further studies using cleaner inhibitors would be necessary to clarify the specific mechanisms involved.

4. The data shown in Fig. 6G, particularly for KG5, is not convincing. There was basically only one concentration point showing difference. More dilutions might help.

Response: this experiment was repeated with additional dilutions as suggested. The data are shown in Fig. R7, and are included in the new Figure 6 in the revised manuscript.

Fig. R7. Effects of KRAS4A of KRAS4B inhibition on sensitivity to KG5.

REVIEWERS' COMMENTS

Reviewer #2 (Remarks to the Author):

The authors submitted a revised manuscript with some additional data and added discussion points. Mainly, a functional validation of the side population as a cancer stem cell population was performed, and additional cell lines from various tissues were analyzed. The response to the reviewer points was thoughtful and thorough and showed a high level of consideration. One major point remains that could not be addressed for reasons discussed by the authors in detail: Because a functional rescue assay of KRAS4A was difficult to interpret, the authors were unable not show that the decreased tumor growth of cells treated with indisulam was a consequence of a reduction of KRAS4A. Furthermore, if inhibition of RBM39 directly causes downregulation of KRAS4A, or if it is a more complex and indirect mechanism, could not be assessed unequivocally. I therefore suggest that the authors refrain from claiming a direct targeting of KRAS4A splicing by inhibition of RBM39, as was done multiple times in the manuscript (Abstract: "Our data identify existing clinical drugs that directly target KRAS4A splicing"; Results: "Our data therefore identify an existing druggable pathway involving inhibition of RBM39 that can directly target expression of the minor KRAS4A isoform."), especially since the authors admit that further analysis is necessary for this claim ("However further detailed analysis by site-specific mutagenesis would be required to verify that RBM39 influences KRAS4A splicing directly rather than through an intermediate protein complex."). Even though this point could not be made, the observations regarding the role of KRAS4A in tumor initiation and its enrichment in cancer stem cell populations, and the inhibition of RBM39 as a potential therapeutic target are interesting, and the evidence and correlations the authors present in this manuscript are relevant to the research community. I therefore conclude that the manuscript is fit to be published, after addressing the above mentioned changes in the text.

Reviewer #3 (Remarks to the Author):

The authors have satisfactorily addressed my concerns

Reviewer #4 (Remarks to the Author):

The authors have responded to all my questions and made the necessary changes to the manuscript. I do not have additional questions and major concerns on the revised version.

Response to reviewer.

The authors submitted a revised manuscript with some additional data and added discussion points. Mainly, a functional validation of the side population as a cancer stem cell population was performed, and additional cell lines from various tissues were analyzed. The response to the reviewer points was thoughtful and thorough and showed a high level of consideration. One major point remains that could not be addressed for reasons discussed by the authors in detail: Because a functional rescue assay of KRAS4A was difficult to interpret, the authors were unable not show that the decreased tumor growth of cells treated with indisulam was a consequence of a reduction of KRAS4A. Furthermore, if inhibition of RBM39 directly causes downregulation of KRAS4A, or if it is a more complex and indirect mechanism, could not be assessed unequivocally. I therefore suggest that the authors refrain from claiming a direct targeting of KRAS4A splicing by inhibition of RBM39, as was done multiple times in the manuscript (Abstract: “Our data identify existing clinical drugs that directly target KRAS4A splicing”; Results: “Our data therefore identify an existing druggable pathway involving inhibition of RBM39 that can directly target expression of the minor KRAS4A isoform.”), especially since the authors admit that further analysis is necessary for this claim (“However further detailed analysis by site-specific mutagenesis would be required to verify that RBM39 influences KRAS4A splicing directly rather than through an intermediate protein complex.”). Even though this point could not be made, the observations regarding the role of KRAS4A in tumor initiation and its enrichment in cancer stem cell populations, and the inhibition of RBM39 as a potential therapeutic target are interesting, and the evidence and correlations the authors present in this manuscript are relevant to the research community. I therefore conclude that the manuscript is fit to be published, after addressing the above mentioned changes in the text.

Response

We thank the reviewer for these positive comments, and agree that we have not yet proven through functional analysis that RBM39 indeed directly binds to Kras mRNA to mediate KRAS4A splicing. We have therefore systematically gone through the manuscript and have removed the words “direct” or “directly” in referring to any interaction between RBM39 and the components of the KRAS pathway.